# DIG In: Evaluating Disparities in Image Generations with Indicators for Geographic Diversity

**Melissa Hall[1], Candace Ross[1], Adina Williams[1], Nicolas Carion[1], Michal Drozdzal[1], Adriana Romero Soriano[1,2,3,4]**

*[1]FAIR, Meta, [2]Mila, Quebec AI Institute, [3]McGill University, [4]Canada CIFAR AI Chair*

**Reviewed on OpenReview:** *https://openreview.net/forum?id=FDt2UGM1Nz*

## Abstract

The unprecedented photorealistic results achieved by recent text-to-image generative systems and their increasing use as plug-and-play content creation solutions make it crucial to understand their potential biases. In this work, we introduce three indicators to evaluate the realism, diversity and prompt-generation consistency of text-to-image generative systems when prompted to generate objects from across the world. Our indicators complement qualitative analysis of the broader impact of such systems by enabling automatic and efficient benchmarking of geographic disparities, an important step towards building responsible visual content creation systems. We use our proposed indicators to analyze potential geographic biases in state-of-the-art visual content creation systems and find that: (1) models have less realism and diversity of generations when prompting for Africa and West Asia than Europe, (2) prompting with geographic information comes at a cost to prompt-consistency and diversity of generated images, and (3) models exhibit more region-level disparities for some objects than others. Perhaps most interestingly, our indicators suggest that progress in image generation quality has come at the cost of real-world geographic representation. Our comprehensive evaluation constitutes a crucial step towards ensuring a positive experience of visual content creation for everyone. Code is available at `https://github.com/facebookresearch/DIG-In/`.

## 1 Introduction

Over the last year, research progress on generative models for visual content creation has experienced a large acceleration. Recent text-to-image generative systems such as Stable Diffusion (Rombach et al., 2021), DALL-E 2 (Ramesh et al., 2022), Imagen (Saharia et al., 2022), and Make-a-Scene (Gafni et al., 2022) have shown unprecedented quality. These systems enable widespread use of plug-and-play content creation solutions, making it crucial to understand their potential biases and out-of-training-distribution behavior.

Prior works in the text and vision domains advocate for detailed documentation of model behavior (Mitchell et al., 2018), introduce benchmarks (Schumann et al., 2021; Hazirbas et al., 2021), and study the potential fairness risks and biases of vision (Buolamwini & Gebru, 2018; DeVries et al., 2019a), language (Bianchi et al., 2022; Smith et al., 2022) and vision-language (Zhao et al., 2017; Hall et al., 2023b) systems trained on large scale crawled data. However, only a handful have focused on better understanding the potential risks and biases of text-to-image generative systems (Cho et al., 2022; Luccioni et al., 2023; Bansal et al., 2022). These works either propose *qualitative* evaluations to highlight biases in generated images or provide narrow quantitative methods focusing on demographic traits of people represented in the images (*e.g.* gender and skintone) and do not take into consideration metrics commonly used in the image generation literature.

Beyond human-centric representations, stereotypical biases may occur for representations of *objects* across the world. Disparities have been observed between continents like Europe, Africa, and the Americas within

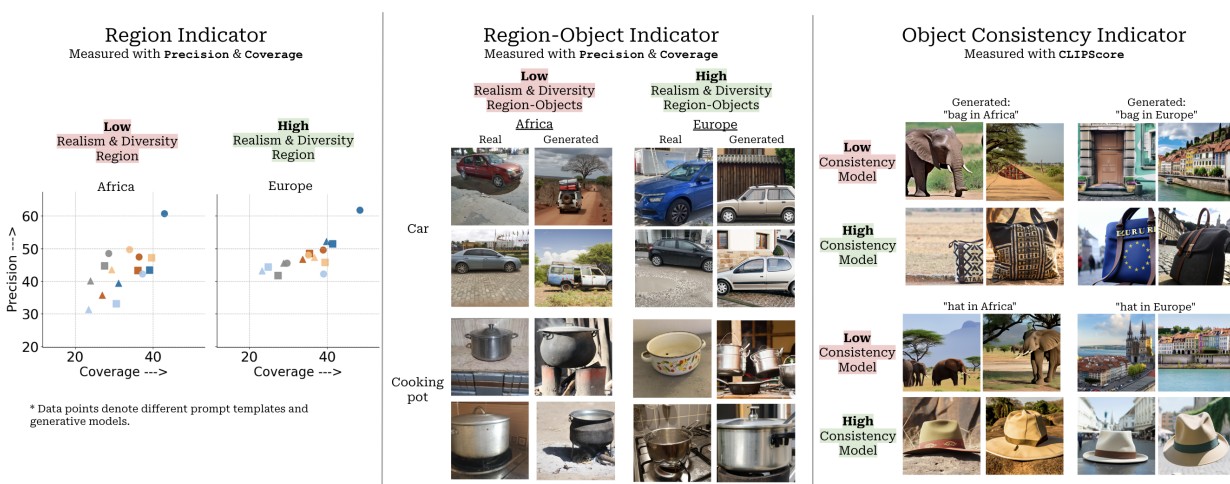

Figure 1: We introduce three quantitative Indicators for measuring gaps in performance between geographic regions in text-to-image models. These Indicators allow for the identification of regions and objects for which state-of-the-art models perform poorly, as shown in the examples here.

third-party object classification systems (DeVries et al., 2019a), weakly- or self-supervised models (Singh et al., 2022; Goyal et al., 2022), and large scale vision-language models (Gustafson et al., 2023). Yet the evaluation of geographic representation of objects and their surroundings in text-to-image systems is under-explored, and mainly utilizes qualitative studies among human annotators to highlight geographic biases in generative text-to-image models.

In this work, we build upon these studies by introducing quantitative benchmarks of geographic disparities in generative models that complement human studies. We introduce three indicators for evaluating the realism, diversity, and consistency of text-to-image generative systems across geographic regions: the **Region Indicator** probes for disparities in realism and diversity of generated images, the **Object-Region Indicator** measures object-specific realism and diversity of generated images, and the **Object Consistency Indicator** evaluates prompt-generation consistency (see Figure 1). The Region and Object-Region indicators are grounded with reference datasets of real images and are adaptable to any real-world dataset of reference images and object prompts.

We then use the indicators to evaluate widely used state-of-the-art text-to-image generative systems. These include two versions of latent diffusion models (Rombach et al., 2021) from free, publicly accessible APIs, denoted as "LDM 1.5 (Open)" and "LDM 2.1 (Open)," and one from a closed, paid API, denoted as "LDM 2.1 (Closed)." We also evaluate a diffusion model utilizing CLIP image embeddings (Ramesh et al., 2022), denoted as "DM w/ CLIP Latents," as well as GLIDE (Nichol et al., 2021). We focus on evaluating whether these systems create content representative of objects' geographical diversity around world regions. We study more than 40 objects using prompts of increasing geographic specificity. For each indicator, we also perform qualitative analyses that demonstrate its efficacy in identifying relevant disparities in generations.

The findings of our analysis are summarized as follows:

- Text-to-image systems tend to generate less realistic and diverse images when prompting with Africa and West Asia than Europe as compared to real, representative images in these regions.

- Prompting with geographic information can reduce the scores of our indicators, highlighting poor object consistency, limited representation diversity, and stereotyped generations.

- Text-to-image systems tend to exhibit region-level disparities more for some objects than others. In some cases, this is due to variations in how objects are represented in the reference dataset of real images, but more often is the result of stereotypes embedded in text-to-image systems.

- Across regions and indicators, the older, publicly accessible latent diffusion model and the newer, closed version tend to be the best and worst performing models, respectively. The DM w/ CLIP Latents model has strong consistency and realism, while GLIDE has strong realism, but suffers from poor diversity and consistency.

- Although these systems have improved in image quality and consistency on standard benchmarks over time (Rombach et al., 2021; Gafni et al., 2022; Ramesh et al., 2022; Saharia et al., 2022), our indicators suggest that this may have been at the expense of real-world geographic representation. For example, the more recent latent diffusion model ranks lower in all three indicators than an earlier version trained with less data.

We hope that our indicators and their use in performing a comprehensive evaluation of sources of geographic bias in generative image models constitute an essential first step towards responsible visual content creation.

## 2 Related work

**Bias and diversity in vision and vision-language systems.** Prior work shows that many societal biases are learned by vision models: race and gender biases are contained in the embedding space of models (Steed & Caliskan, 2021; Wolfe & Caliskan, 2022b), and disparities for different demographic groups occur in tasks such as image classification (Goyal et al., 2022; Hall et al., 2023a) and object detection (DeVries et al., 2019a; Gustafson et al., 2023). Similar to vision models, vision-language models, *e.g.* CLIP (Radford et al., 2021), contain harmful biases in their embedding spaces (Ross et al., 2021; Srinivasan & Bisk, 2021; Wolfe & Caliskan, 2022a). Models also show biases on downstream tasks including image captioning (Hendricks et al., 2018) and video understanding (Hazirbas et al., 2021; Porgali et al., 2023). In addition, these analyses extend to the large-scale datasets used for training and benchmarking (Birhane et al., 2021; Hazirbas et al., 2021; Porgali et al., 2023; Ramaswamy et al., 2023). These works tend to focus on demographic biases related to visual person-related attributes, with less attention on geographic biases.

**Bias and diversity in generative text-to-image systems.** Recent papers analyzing biases in generative models (Luccioni et al., 2023; Friedrich et al., 2023; Cho et al., 2022; Bansal et al., 2022) utilize automated metrics to identify biases in image generative systems but primarily focus on disparities across gender and ethnicity of groups of people rather than geographically diverse representations of objects. Bianchi et al. (2022) study both demographic biases and geographic diversity in text-to-image systems *qualitatively*, showcasing the presence of bias in text-to-image systems. Concurrent work (Basu et al., 2023) measures the geographic representation of objects in text-to-image systems by performing user studies on a subset of 10 objects. None of these works performs a *systematic quantitative analysis* of object geodiversity of text-to-image systems, grounds the analysis on a reference dataset, or considers a large scale number of objects and their associated generations.

**Generative models metrics.** The evaluation of conditional image generative models often involves different metrics to assess desirable properties of the models, such as image quality/realism, generation diversity, and conditioning-generation consistency (DeVries et al., 2019b). Fréchet Inception Distance (FID; Heusel et al. 2017b) and Inception Score (Salimans et al., 2016) have been extensively used to measure image generation quality and diversity. These metrics yield distribution-specific scores grouping quality and diversity aspects of conditional image generation, therefore hindering the analyses of trade-offs and individual failures cases. To overcome these limitations, precision and recall-based metrics (Sajjadi et al., 2018; Kynkäänniemi et al., 2019; Naeem et al., 2020; Shmelkov et al., 2018; Ravuri & Vinyals, 2019) disentangle visual sample quality/realism and diversity into two different metrics. The diversity of generated samples has also been measured by computing the perceptual similarity between generated image patches (Zhang et al., 2018) or, more recently, the Vendi score (Friedman & Dieng, 2022). In addition, consistency of generated images with respect to the input conditioning has been assessed by leveraging modality specific metrics and pre-trained models. For example, the CLIPscore (Hessel et al., 2021) computes the cosine similarity between text prompts used in text-to-image generative models and their corresponding generations, and TIFA (Hu et al., 2023) assesses consistency through visual question answering. In this work, we introduce indicators

| Model | Dataset | {object} | Prompt {object} in {region} | {object} in {country} |
|---|---|---|---|---|
| LDM 1.5 (Open) LDM 2.1 (Open) LDM 2.1 (Closed) GLIDE | GeoDE | Object-region Balanced (29K) | Object-region Balanced (29K) | Object-region Balanced (29K) |
| | DollarStreet | Full (22K) | Full (22K) | Full (22K) |
| DM w/ CLIP Latents | GeoDE | Object-region Balanced (29K) | Object-region Balanced (29K) | ✗ |
| | DollarStreet | ✗ | ✗ | ✗ |

Table 1: Number of samples generated per dataset, for each model and prompt. **Full** indicates the original distribution of the corresponding dataset. **Object-region Balanced** denotes a fixed number of images for each object and region. Using a balanced dataset controls correlations between regions and objects.

that leverage precision, coverage, and CLIPscore to report measurements *disaggregated by geographic region*, allowing for a granular understanding of potential disparities of text-to-image generative models.

## 3 Experimental Set-up

The proposed indicators require access to reference datasets of real images, a pre-trained image feature extractor, and pre-trained text-to-image systems which are queried to gather datasets of generated images. We introduce the datasets, models, and prompts used in this Section and describe the Indicators in Section 4.

### 3.1 Reference, real world datasets

We use GeoDE (Ramaswamy et al., 2023) and DollarStreet (Rojas et al., 2022) as reference datasets to compute realism and diversity of the generated images. Example images are shown in Appendix A.1.

GeoDE contains images of 40 objects classes across the geographic regions of Africa, the Americas, West Asia, Southeast Asia, East Asia, and Europe. Images were taken by residents of each region and are subject to strict quality requirements ensuring objects fill at least 25% of the image and are not occluded or blurred.

DollarStreet contains images of approximately 200 concept classes across Africa, the Americas, Asia, and Europe. Professional and volunteer photographers especially targeted "poor and remote environments" (Gapminder, 2021) to take photos of relevant objects and scenes. We filter out images containing multiple object labels.

We filter each dataset to include only classes that are objects and have images for all regions. We exclude subjective classes (*e.g. nicest shoes* or *favorite home decorations*). While GeoDE is roughly balanced across regions and objects by design, we sample to ensure the same number of images for each object and region to control for any correlations. This yields 27 objects with 180 images per object-region combination. DollarStreet is quite imbalanced, so we use the original distribution to maintain a sufficient number of samples for each region. This yields 95 objects with approximately 3,500 images from Africa, 3,500 images from the Americas, 9,000 images from Asia, and 2,700 images from Europe. In order to use the image feature extractor required to compute the proposed geodiversity indicators, we center crop all images.

### 3.2 Models

We evaluate generated images across five different models: two from commercial APIs and three from open source APIs. To maintain consistency, all image resolutions are $512 \times 512$. Where possible, we use the safety filter associated with each system. We find that the filter affects approximately 0.3% of generated images, roughly balanced across region and prompt. All models were accessed in May and June of 2023.

We first evaluate several versions of latent diffusion models (Rombach et al., 2021). The first, which we term "LDM 1.5 (Open)," is trained on a public dataset of approximately 2 billion images followed by further steps on higher resolution images and fine-tuning on aesthetic images. We access the model via an API containing open-sourced model weights. The second, "LDM 2.1 (Open)," is initially trained on a public dataset of approximately 5 billion images excluding explicit material, further trained for multiple iterations on increasingly high-resolution samples, then fine-tuned. We also access this model via an API containing open-sourced model weights. The third, "LDM 2.1 (Closed)," is similar to the aforementioned model but accessed via a paid API intended for artistic creativity. We follow the recommended parameters for each system, and sample with 30 diffusion steps for LDM 2.1 (Closed) and with 50 steps for the open sourced API. We use a classifier-free guidance scale of 7.[1]

We also evaluate GLIDE (Nichol et al., 2021), an open-sourced diffusion model using classifier-free guidance trained on a dataset filtered to remove images of people and violent objects. We follow the suggested parameters and sample with 100 diffusion steps. Finally, we evaluate a model that uses a multi-modal implementation of a generative pre-trained transformer leveraging CLIP image embeddings (Ramesh et al., 2022) and has approximately 3.5 billion parameters. We call this model "DM w/ CLIP Latents," and access it through a paid API.

### 3.3 Querying text-to-image generative systems to obtain datasets of generated images

We query text-to-image systems with prompts of varying geographical specificity. First, we prompt with `{object}`, referring to any object class name in the reference dataset. Second, we prompt with `{object} in {region}`, using the region defined in the real world dataset. Third, to further understand how the specificity of the geographic prompting affects the image generation, we use the prompt `{object} in {country}`. For each region and object combination in GeoDE, we balance the prompt across the top three countries for the respective region included in the reference dataset. For DollarStreet, we use the most represented countries per region.

In Table 1 we show the quantity of images generated for each prompt. In our generations, we match the balanced version of GeoDE, consisting of 27 objects each represented by 180 images per region. For DollarStreet, we follow the original distribution of images of the reference dataset.

## 4 Geodiversity indicators

In this section, we first discuss relevant metrics used in the image generation literature and how they map to realism (also referred to as quality), diversity, and consistency of generated images. Then, we introduce geodiversity indicators that build on these metrics to quantitatively evaluate disparities across regions.

### 4.1 Realism, diversity and consistency metrics

Rather than using a single metric like FID (Heusel et al., 2017b) to evaluate disparities, we disaggregate across several metrics in order to better understand possible trade-offs between realism, diversity, and consistency.

**Realism (Precision):** We measure realism by quantifying how similar generated images are to a reference dataset of real images using precision. We follow the definition of precision introduced in (Kynkäänniemi et al., 2019), which determines the proportion of generated images that lie within a manifold of real images. The manifold of real images is estimated by building hyperspheres around data points of real images, where the radii of hyperspheres is controlled by the distance to the $k$-th nearest neighbor in a pre-defined feature space, *e.g.*, the feature space of the Inceptionv3 (Szegedy et al., 2015). Formally, precision is computed as:

$$P(\mathcal{D}_r, \mathcal{D}_g) = \frac{1}{|\mathcal{D}_g|} \sum_{i=1}^{|\mathcal{D}_g|} \mathbb{1}_{h_g^{(i)} \in \text{manifold}(\mathcal{D}_r)}, \tag{1}$$

---

[1]We note that increasing the number of steps does not appear to alter the trends observed in this analysis. Similarly, increasing the classifier-guidance does not result in any significant improvements of the observed consistency shortcomings.

where $\mathcal{D}_r = \{h_r^{(j)}\}$ is a dataset of real image features, and $\mathcal{D}_g = \{h_g^{(i)}\}$ is a dataset of generated image features.

It is worth noting that, in the case of geodiversity indicators, we aim to account for variations in the distribution of reference images across geographic region. For example, regions with more economic diversity may have larger manifolds of images due to more variation in objects related to economic status, such as houses and cars. As a result, improved metrics that measure realism in terms of density may not be robust to these variations, as density more strongly rewards generated images whose features occur in regions densely packed with real samples.

**Diversity (Coverage):** To measure diversity of generated images, we use coverage (Naeem et al., 2020). As in the case of precision, coverage requires estimating the manifold of real images from a reference dataset. Coverage is measured as the proportion of real-data hyperspheres that contain generated images and is more robust to outliers than recall – the metric that is usually used to complement precision. Because it measures the extent to which real data is represented by the generated data, coverage indicates the generated images' diversity. Formally, coverage is computed as:

$$\mathrm{C}(\mathcal{D}_r, \mathcal{D}_g) = \frac{1}{|\mathcal{D}_r|} \sum_{j=1}^{|\mathcal{D}_r|} \mathbb{1}_{\exists\, i \text{ s.t. } h_g^{(i)} \in \text{hypersphere}\left(h_r^{(j)}\right)}. \tag{2}$$

It is worth noting that some text-to-image systems generate unrealistic – *e.g.* low quality or inconsistent – yet diverse samples, thus metrics that rely on estimating the manifold of generated images, such as recall (Kynkäänniemi et al., 2019), may lead to manifold over-estimations which in turn may result in unreliably high measured diversities.

**Consistency (CLIPScore):** We measure the input-output consistency of the generative model by computing the CLIPscore (Hessel et al., 2021) – *i.e.*, cosine similarity – between two embeddings. The first embedding corresponds to the text prompt CLIP embedding used to condition the text-to-image generation process, whereas the second embedding corresponds to the CLIP embedding of the generated image. Intuitively, the higher the CLIPscore, the higher the input-output consistency. Formally, CLIPscore is defined as:

$$\mathrm{CLIPscore}\left(x_g^{(i)}, p^{(i)}\right) = \frac{f_v\left(x_g^{(i)}\right) \cdot f_t\left(p^{(i)}\right)}{||f_v\left(x_g^{(i)}\right)||\, ||f_t\left(p^{(i)}\right)||} \tag{3}$$

where $x_g^{(i)}$ is an image generated with the prompt $p^{(i)}$, $f_v$ is the visual CLIP embedding function, and $f_t$ is the textual CLIP embedding function.

### 4.2 Region Indicator

**Goal.** The Region Indicator seeks to probe text-to-image generative systems for disparities in the realism and diversity of generated images across different regions.

**Requirements.** The Region Indicator requires a reference dataset of real images containing images from around the world and associated regional labels, a generated dataset of images from a pre-trained text-to-image generative system, and a pre-trained image feature extractor – we use the InceptionV3 (Szegedy et al., 2015) given its ubiquity in the literature to compute metrics such as FID (Heusel et al., 2017a; Casanova et al., 2021; Ramesh et al., 2022; Rombach et al., 2021).

For reference datasets, we split GeoDE into 6 regional datasets each containing images of objects from non-overlapping regions in the original dataset. We split DollarStreet into 4 regional datasets each containing images of objects from non-overlapping regions in the original dataset.

For generated datasets, we consider three sets of generated images obtained using the `{object}`, `{object} in {region}`, and `{object} in {country}` prompts. Following the reference datasets, we split each of the

generated datasets into regional datasets containing generated region-specific images. For the `{object} in {region}` prompt, we consider all object images generated for a given region to constitute each regional dataset. For the `{object} in {country}` prompt, we compose regional datasets by merging generations for all countries within the same region, where the country list is defined by the GeoDE and DollarStreet datasets. For the `{object}` prompt, we sample the image generations such that the object distribution matches the one in the reference region-specific datasets.

**Metrics.** We compute precision and coverage per region, where each region is represented by the set of objects in the reference datasets. In particular, we compute $P(\mathcal{D}_r^R, \mathcal{D}_g^R)$, where $\mathcal{D}_r^R$ is the dataset of features of real images from region $R$, and $\mathcal{D}_g^R$ is the generated dataset of features of generated images associated with region $R$. In the same spirit, we compute $C(\mathcal{D}_r^R, \mathcal{D}_g^R)$. This coverage definition measures the overall regional diversity by grouping object-specific conditionings according to region.

### 4.3 Object-Region Indicator

**Goal.** The Object-Region Indicator seeks to probe text-to-image generative system by measuring object-specific realism and diversity of generated images of objects across different regions, complementing the Region Indicator by providing detailed information at object level.

**Requirements.** The Object-Region Indicator requires a reference dataset of real images containing images of objects from around the world and their associated object class and regional labels, a generated dataset of images from a pre-trained text-to-image generative system, and a pre-trained image feature extractor – once again, we use the InceptionV3.

For reference datasets, we split GeoDE into every possible object-region combination present in the original GeoDE dataset. We do not use the DollarStreet dataset as it is very imbalanced and the number of samples for certain object-region combinations is too small.

For generated datasets, we consider three datasets of generated images, obtained using the `{object}`, `{object} in {region}`, and `{object} in {country}` prompts. Following the reference datasets, we split the generated datasets into as many datasets as object-region combinations present in the GeoDE dataset. As in Section 4.2, we merge `{object} in {country}` generations for all countries within the same region and sample the `{object}` generations to follow the reference distributions of object-region images.

**Metrics.** We compute precision and coverage per object-region combination. In particular, we compute $P(\mathcal{D}_r^{O,R}, \mathcal{D}_g^{O,R})$, where $\mathcal{D}_r^{O,R}$ is the dataset of features of real images of object $O$ in region $R$, and $\mathcal{D}_g^{O,R}$ is the dataset of features of generated images corresponding to object $O$ and region $R$. Similarly, we compute $C(\mathcal{D}_r^{O,R}, \mathcal{D}_g^{O,R})$. Note that, contrary to the region indicator, the object-region coverage measures object-region conditional diversity – *i.e.* given an object-region prompt, how diverse the generations are.

### 4.4 Object Consistency Indicator

**Goal.** The Object Consistency Indicator seeks to probe text-to-image generative systems for disparities in consistency across generations of specific objects from around the world.

**Requirements.** By contrast to other indicators, the Object Consistency Indicator does not rely on a reference dataset of images. Instead, it requires a dataset of object-prompts, a generated dataset of images from a pre-trained text-to-image generative system, and pre-trained image and text feature extractors (CLIP).

For object prompts, we build a dataset of prompts `{object}` featuring all objects in the GeoDE dataset.

For generated datasets, we consider three sets of generated images, obtained using the `{object}`, `{object} in {region}`, and `{object} in {country}` prompts. We split each of the generated datasets into as many datasets as object-region combinations present in the GeoDE dataset. We follow the procedure described in sections 4.2 and 4.3 to compose region-specific datasets of generated images.

**Metrics.** For each generation, we compute the CLIPScore between the generated images and the `{object}` prompt as follows: $\text{CLIPscore}\left(x_g^{(i)}, p^{(i)}\right)$, where $x_g^{(i)}$ is the generated image obtained by conditioning the text-to-image system with prompt $p^{(i)}$. Next, we stratify the scores by object. For each object class, we compute the 10th percentile CLIPScore[2]. We report the 10th-percentile CLIPScore across all objects as the consistency indicator.

## 5 Results and observations

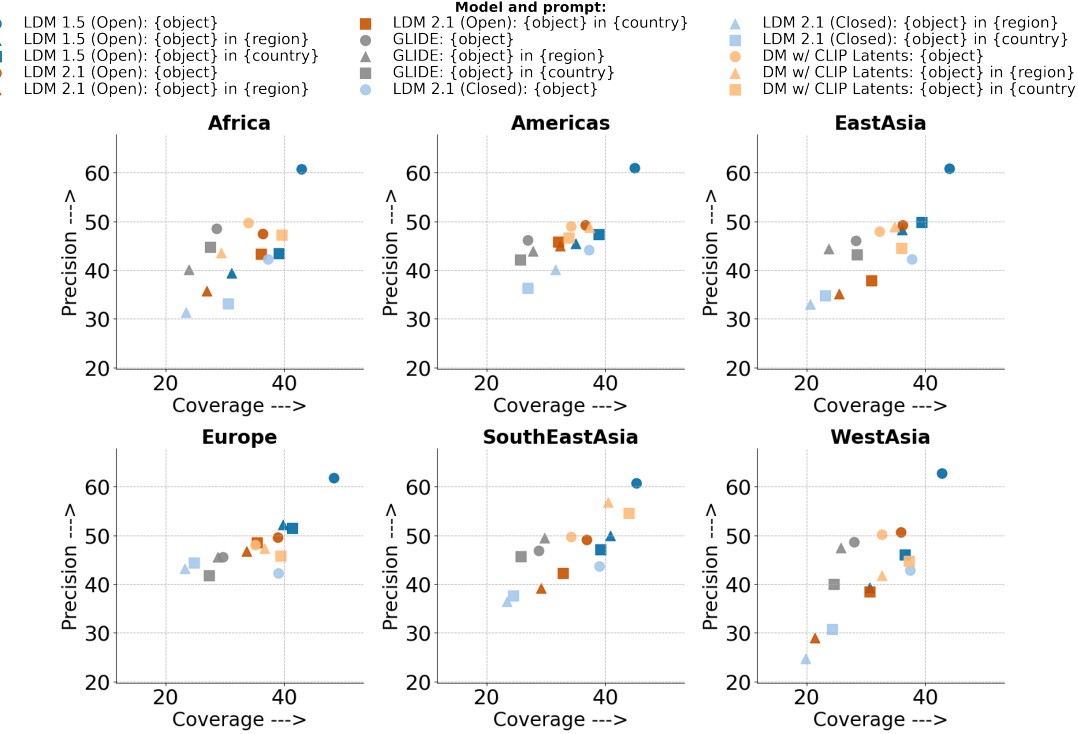

Figure 2: Precision (quality) and coverage (diversity) measurements evaluated with the GeoDE dataset.

We now present quantitative results for the three indicators and perform qualitative analyses for some exemplar regions and objects. Additional qualitative examples are included in the Appendix. For each indicator, we first discuss region-level disparities, then trends that occur for all regions.

### 5.1 Region Indicator

Figure 2 show results for the Region Indicator using GeoDE as the reference dataset, and results with DollarStreet are shown in Appendix A.5.

**Disparities when prompting without geographic information.** For the `{object}` prompt, precision and coverage are quite similar across regions. As an example, the first column of Figures 3 and 4 show images for *car* and *stove*. Indeed we see that, for a given model, the generated images are not especially similar to a specific region as compared to the others.

---

[2]By looking at the distribution of CLIPscores, we observe that the spread of the distribution, and in particular, the tail of the distribution corresponding to the lowest CLIPscores, tend to be the most informative to capture inconsistencies in the generations. This is perhaps unsurprising given how CLIPscore has been used as a filtering tool to display the best image generations of a system (He et al., 2022).

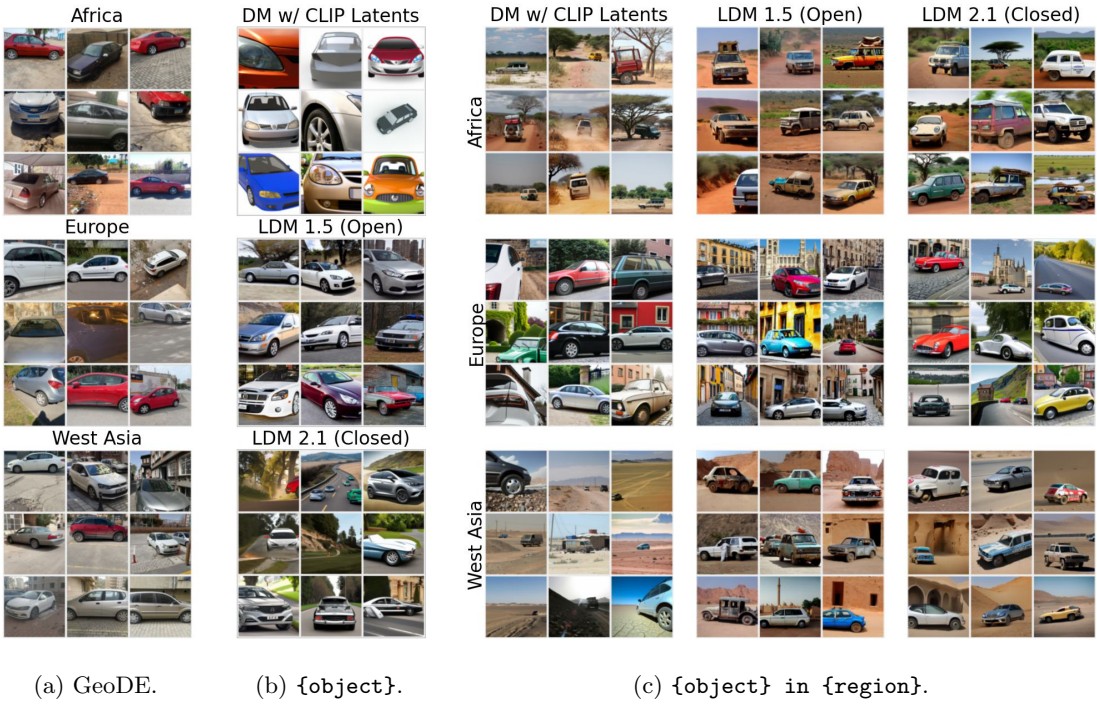

(a) GeoDE.    (b) {object}.    (c) {object} in {region}.

Figure 3: Random examples of *cars*.    **Sub-figure (a)** Real images from Africa, Europe and West Asia. **Sub-figure (b)** Generations obtained with {object} setup using three models. **Sub-figure (c)** Generations obtained with {object} in {region} for three models and three regions.

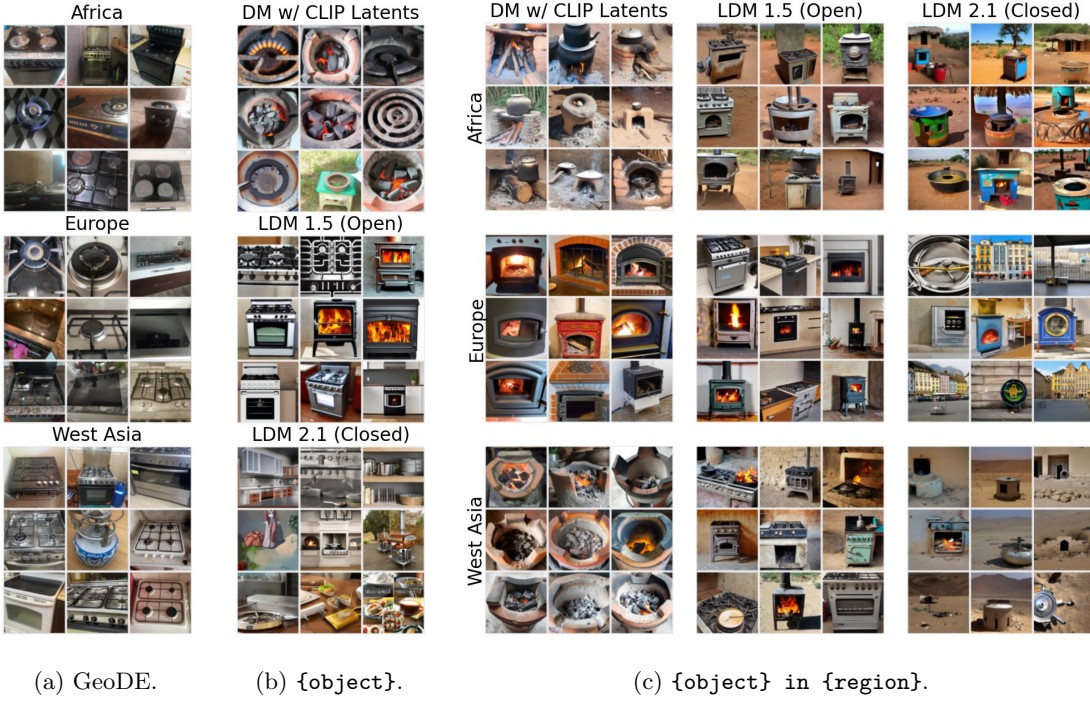

(a) GeoDE.    (b) {object}.    (c) {object} in {region}.

Figure 4: Random examples of *stoves*.    **Sub-figure (a)** Real images from Africa, Europe and West Asia. **Sub-figure (b)** Generations obtained with {object} setup using three models. **Sub-figure (c)** Generations obtained with {object} in {region} for three models and three regions.

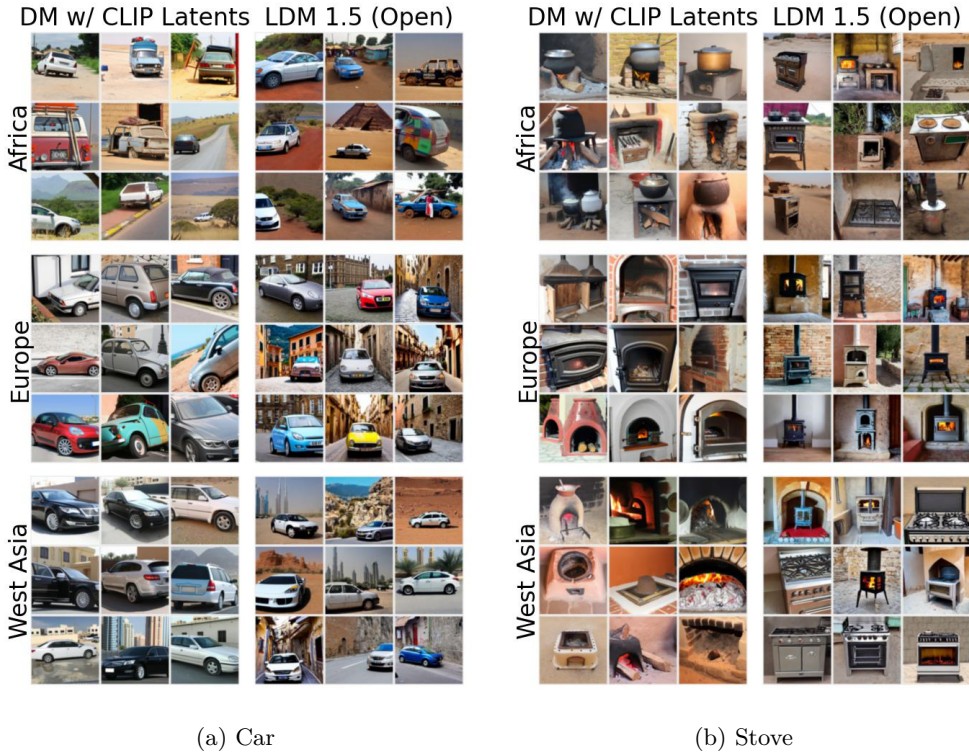

(a) Car                                           (b) Stove

Figure 5: Random examples of generations of *cars* and *stoves* obtained with `{object} in {country}` setup.

**Disparities when prompting with region.**   When prompting with region, patterns of disparities appear. When evaluating with GeoDE as the reference dataset, generations corresponding to Africa, East Asia, and West Asia tend to have poorer coverage and lower precision than Europe and the Americas. Upon visual inspection, generations using region prompting tend to be homogeneous and stereotyped for lower performing regions and do not capture the diversity of the objects in the real dataset. For example, generated images of *cars* prompted for Africa (Figure 3) are almost exclusively boxy SUVs in rural or desert-like backgrounds, which are rare in the real dataset. Generated images of *stoves* in West Asia (Figure 4) are circular basins filled with coal (for DM w/ CLIP Latents) or stove-oven combinations in outdoor settings (for LDM 1.5 (Open)), neither of which appear often in the real dataset.

**Disparities when prompting with country.**   We find that the coverage increases for all models except GLIDE, and most regions, when prompting with country names rather than regions. This generally comes at little cost to, and in some cases helps, precision. Visual inspection confirms that prompting with country rather than region tends to yield more diverse generations. For example, the images of *stoves* generated by DM w/ CLIP Latents and LDM 1.5 (Open) for the region prompt (Figure 4) are more homogeneous for a given region than when prompted with `{object} in {country}` (Figure 5). Similarly, the background of images showing *cleaning equipment* show both indoor and outdoor scenes across all regions when prompting with country, while generally depicting only one or the other for the region prompt. For GLIDE, prompting with country information has a mixed effect, increasing coverage for Africa and East Asia but decreasing coverage and precision for all other regions.

**Observations consistent across regions.**   The more recent LDM 2.1 (Closed) and LDM 2.1 (Open) tend to have lower precision and coverage than the older LDM 1.5 (Open). This could be due to extra rounds of aesthetic training in latter versions of the models, causing greater deviation from real-world images. In addition, all models except DM w/ CLIP Latents have a degradation in precision and coverage when including region information in prompts. We suspect this is because prompting with geography causes stereotypical

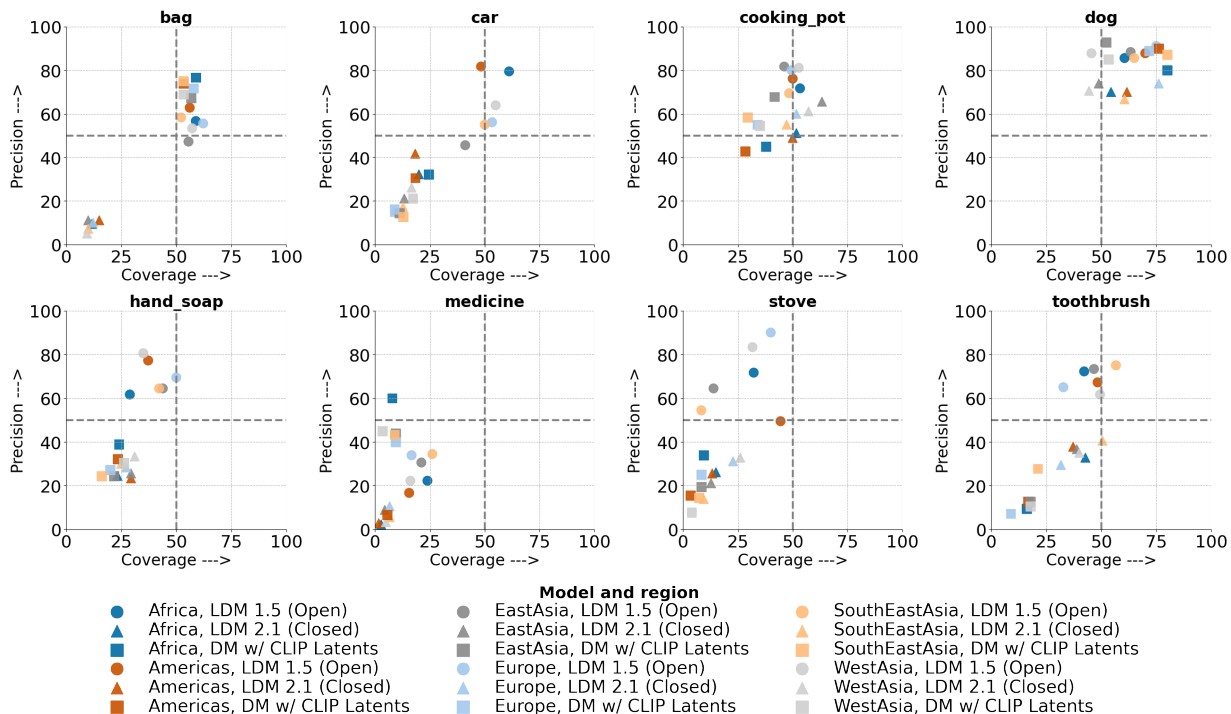

Figure 6: Region-Object Indicator for `{object}` prompt.

backgrounds and objects that are generally not represented in the real, reference dataset. Our DollarStreet results provide additional insights in this regard: while our previously observed patterns across models and prompts are generally consistent when evaluating with DollarStreet, we find that prompting with regions shows an *improvement* for Africa (shown in Appendix A.5). This reversal in trend for Africa between GeoDE and Dollar Street may be due to Dollar Street's focus on lower-income, less photographed locations, as well as its lack of enforcement of a minimum size for pictured objects. Its images likely contain lower-income settings and pronounced backgrounds more similar to the generated images depicting stereotypes of the region.

## 5.2 Region-Object Indicator

Figures 6–8 show precision and coverage results for the Object-Region Indicator using eight objects found in GeoDE for DM w/ CLIP Latents, LDM 2.1 (Closed) and LDM 1.5 (Open). We focus on these objects in particular as they highlight notable patterns of disparities that can be identified with this Indicator. Additional results covering all models and objects can be found in the Appendix.

**Disparities when prompting without geographic information.** When interpreting the `{object}` prompt results in Figure 6, we observe that precision and coverage patterns vary across objects. First, objects such as *bag* exhibit rather consistent precision and coverage across regions, but vary across models. The low variance in the inter-region measurements suggest that – given the generated images remain constant across comparisons – *bag* is similarly represented across regions in the reference dataset. Figure 9 shows that this is true for GeoDE. While generations of DM w/ CLIP Latents and LDM 1.5 (Open) depict bags, those of LDM 2.1 (Closed) align with the model's lower precision and coverage in rarely showing bags.

Second, objects have more variation in coverage across regions than in precision for all models. For example, the concept *dog* tends to have lower coverage in West Asia than other regions, whereas coverage for Europe tends to be higher. This suggests there are differences in representation of the concept *dog* in the reference dataset between these regions. In Figure 10, we observe that dogs lying by grass and photographed from above appear more frequently for West Asia in GeoDE and are not common in the generations.

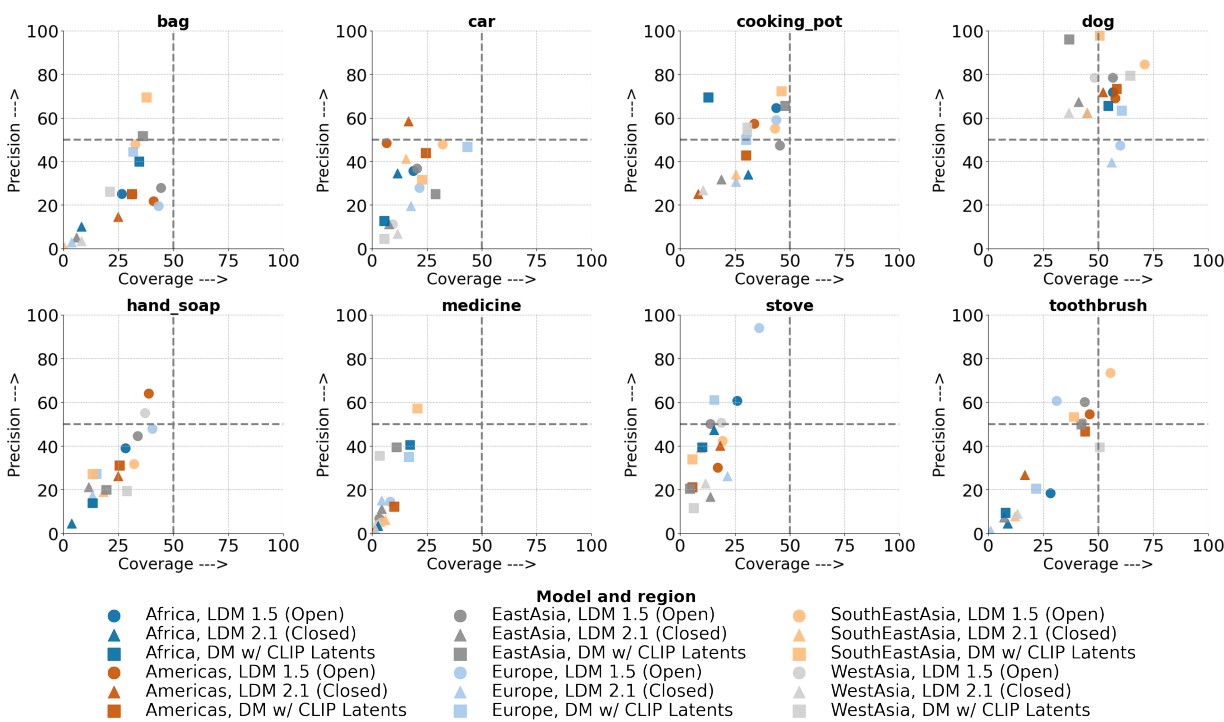

Figure 7: Region-Object Indicator for `{object}` in `{region}` prompt.

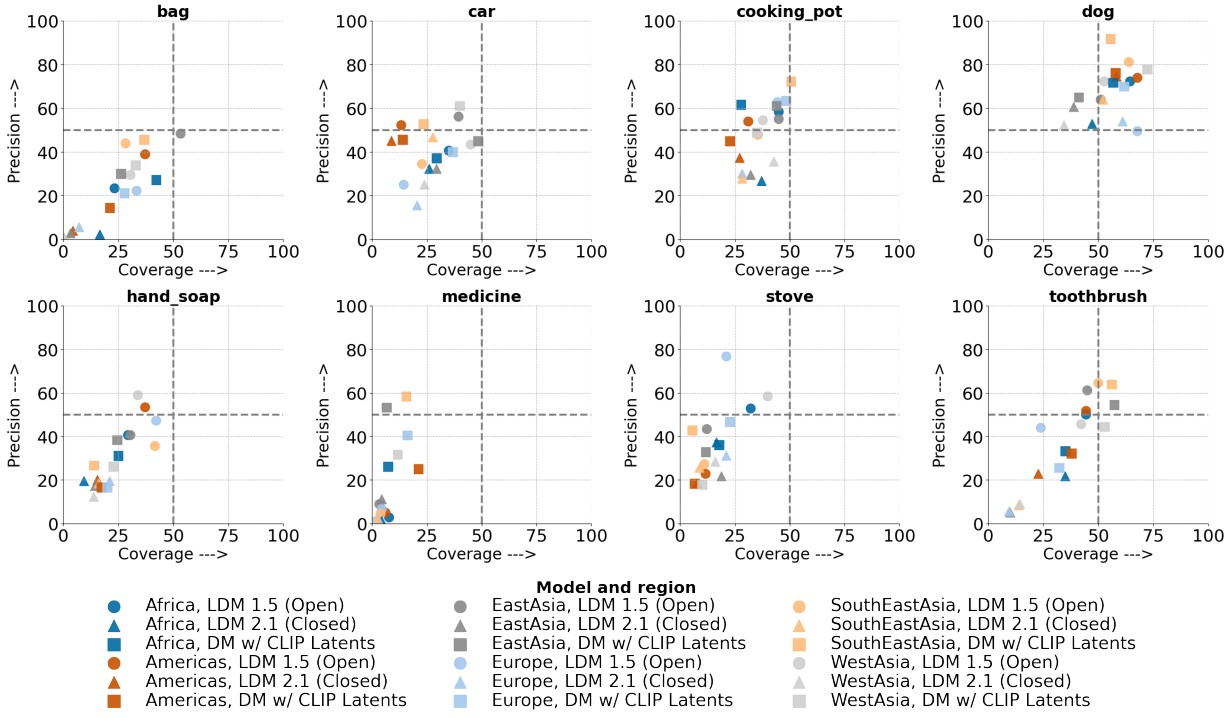

Figure 8: Region-Object Indicator for `{object}` in `{country}` prompt.

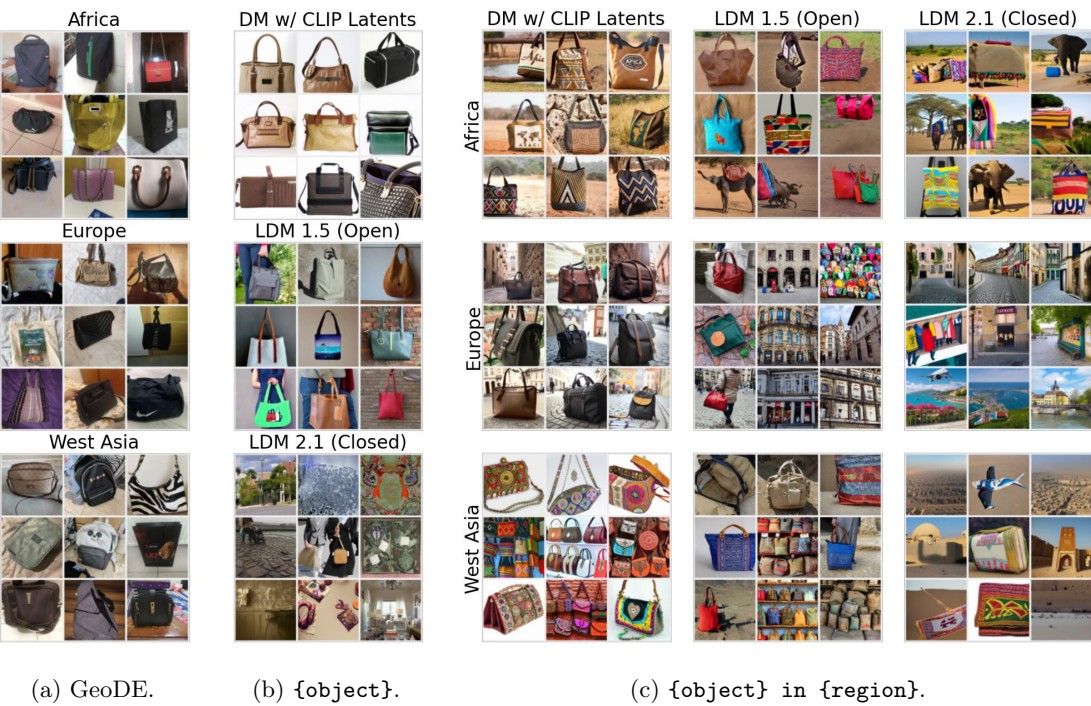

(a) GeoDE.   (b) {object}.   (c) {object} in {region}.

Figure 9: Random examples of *bags*. **Sub-figure (a)** Real images from Africa, Europe and West Asia. **Sub-figure (b)** Generations obtained with {object} setup using three models. **Sub-figure (c)** Generations obtained with {object} in {region} for three models and three regions.

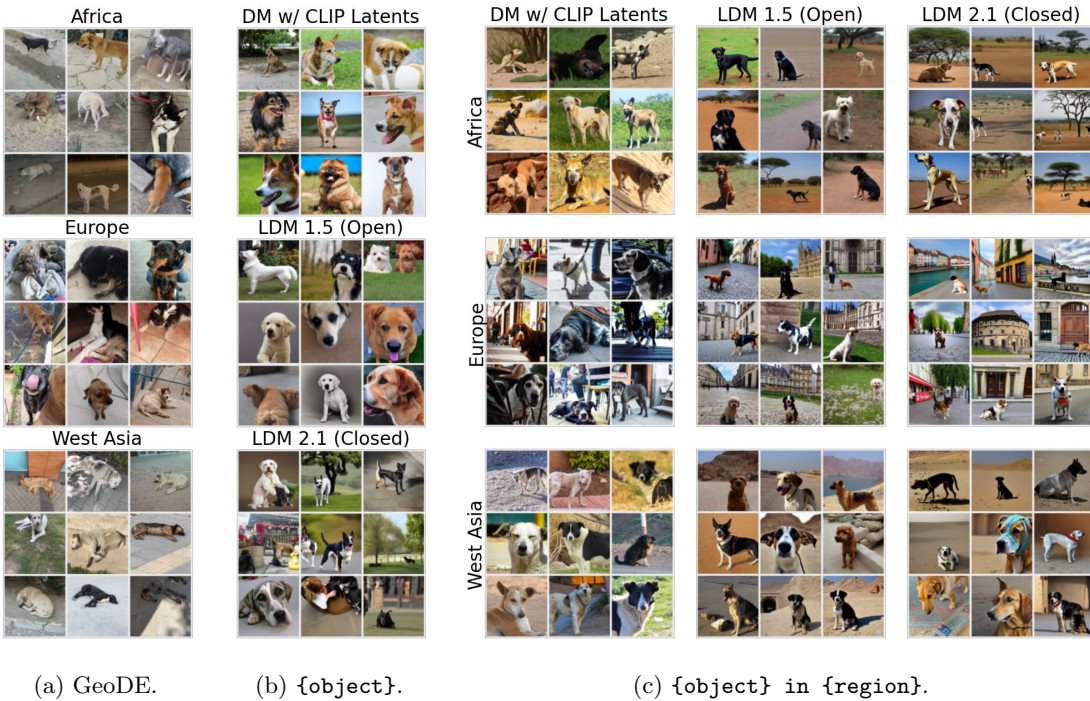

(a) GeoDE.   (b) {object}.   (c) {object} in {region}.

Figure 10: Random examples of *dogs*. **Sub-figure (a)** Real images from Africa, Europe and West Asia. **Sub-figure (b)** Generations obtained with {object} setup using three models. **Sub-figure (c)** Generations obtained with {object} in {region} for three models and three regions.

Third, objects such as *cooking pot*, *hand soap*, and *stove* have notable disparities in precision and coverage across regions for at least one model. For example, *cooking pot* shows stronger precision in East Asia than Africa and the Americas, despite the Americas and Africa showing higher coverage for LDM 1.5 (Open). For *hand soap*, Africa achieves among the top precision and coverage for DM w/ CLIP Latents and Southeast Asia the least. The disparities observed for the concept *stove* may be explained by the mismatch between representation of the concept in the generations versus different regions of the GeoDE dataset. Figure 4 shows that LDM 1.5 (Open)'s generations of *stoves* focus primarily on frontal perspectives of freestanding stoves/fireplaces. Yet, these generations do not include portable stoves nor rectangular gas cooktops found in GeoDE's West Asia, aligning with the lower coverage values. DM w/ CLIP Latents's generations of the concept *stove* depict close-ups of single burners with fire lit up in many cases, which do not appear frequently in GeoDE. LDM 2.1 (Closed) does not always display stoves when prompted for this object.

**Disparities when prompting with region.** When considering the `{object} in {region}` prompt results in Figure 7, some objects like *medicine* continue to show disparities similar to those for the `{object}` prompt while other trends change. This is unsurprising given that the generative systems now leverage specific region information. For example, objects such as *bag* and *toothbrush* now present accentuated performance disparities – more notably in precision, but also in coverage – for DM w/ CLIP Latents and LDM 1.5 (Open). These new trends are captured in Figure 9, which shows that these two models most often generate bags outdoors surrounded by rather stereotyped backgrounds. For example, backgrounds in generations of Europe include pavements made of setts or cobblestones, whereas those in Africa more often depict rural areas. For *stove*, coverage becomes low for most models and regions, and precision displays disparities of up to 70% within a single model. Figure 4 shows that DM w/ CLIP Latents and LDM 1.5 (Open) generate interior freestanding stoves/fireplaces, often with lit fires, for Europe and predominantly outdoor rudimentary stoves for West Asia and Africa. Few of these representations are common in GeoDE.

As seen for the Region Indicator, including region information when prompting harms the performance across regions and models for many object classes. This includes *dog*, which previously achieved among the highest precision and coverage scores. Figure 10 highlights that dogs tend to appear outdoors when region information is included, and, similar to *bag* generations, the image backgrounds appear rather stereotyped.

We find that the prompt specificity affects the LDM 2.1 (Closed) model particularly, resulting in lower precision and coverage than other systems for many objects including *medicine*, *bag*, *cooking pot*, or *toothbrush*. Figures 4 and 9 show that LDM 2.1 (Closed) appears to disregard the object to be generated and produces images of outdoor scenes that likely represent the model's perception of each region. We investigate this more with the Object Consistency Indicator in the following section.

**Disparities when prompting with country.** When focusing on `{object} in {country}`-prompted generations in Figure 8, we continue to observe disparities in performance across regions for all models. Intuitively, given the broad regions considered, providing country-specific information may guide the model towards different generations than when providing region information. Prompting with country appears to improve the performance for some object classes, but hurt it for others. For example, adding country-specific information appears to benefit the generations of *car* in many regions. Figure 5 shows that DM w/ CLIP Latents and LDM 1.5 (Open) now exhibit more diversity of car types in Africa – including both SUVs and sedans – and car surroundings – still dominated by rural areas but now with more infrastructure. Car generations of countries in West Asia show increased diversity in car types as well as an increased presence of cars in urban areas. Similarly, in the case of *stove*, prompting LDM 1.5 (Open) with countries within West Asia leads to generations which include freestanding electric and gas stoves. These observations are echoed in Figure 8, which highlights an increase in precision and coverage for those models, objects, and regions.

When considering all 27 object classes, we observe that making the prompt more specific (from regions to countries) helps improve precision and coverage on average for Africa, West Asia and East Asia for most models (see Figure 16 in Appendix).

**Observations consistent across regions.** For `{object}`, LDM 1.5 (Open) achieves the overall highest object-region performance, with only a few objects below a 50% performance threshold for both precision

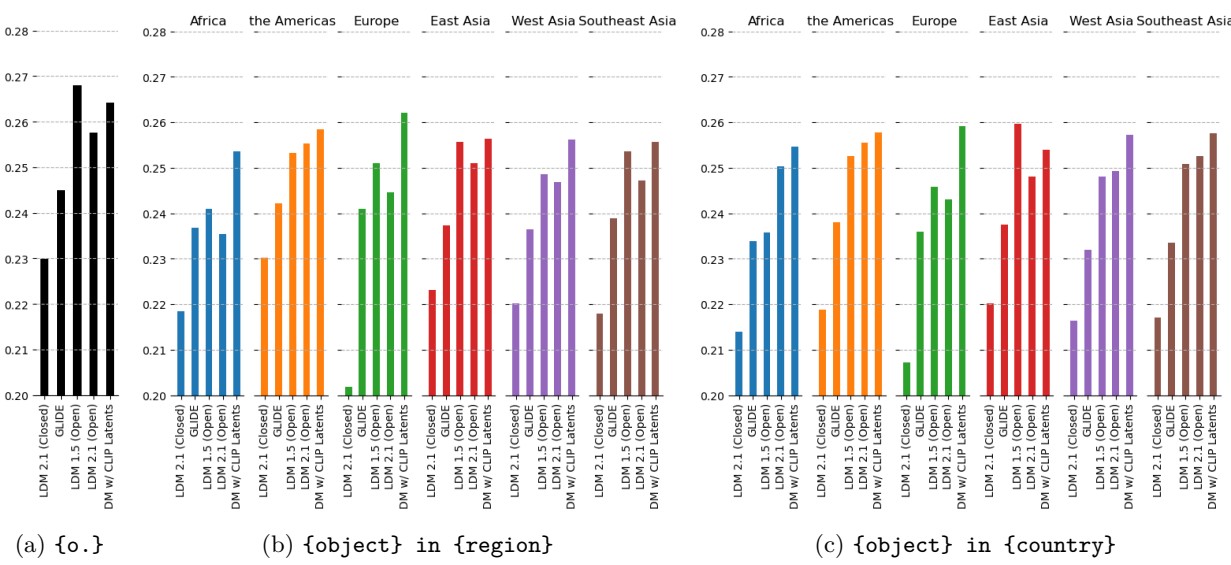

(a) {o.}  (b) {object} in {region}  (c) {object} in {country}

Figure 11: Geode dataset prompts, bar plots representing average of per class lowest 10% Clipscores.

and coverage. Although for some objects such as *bag*, DM w/ CLIP Latents shows better precision-coverage results, the higher precision values often come at the expense of lower coverage – see *e.g. candle* or *dog*. For {object} in {region}, LDM 2.1 (Closed) exhibits low performance (below 30% in both precision and coverage) for a large number of object-region combinations. The poor performance of LDM 2.1 (Closed) may highlight the overall lack of consistency in the system's generations perceived upon visual inspection. Yet, we observe that for {object} in {country}, the low performance previously observed for some objects slightly recovers. Overall, LDM 1.5 (Open) appears to benefit more from making the prompts more specific than its paid-API counterpart LDM 2.1 (Closed).

## 5.3 Object Consistency Indicator

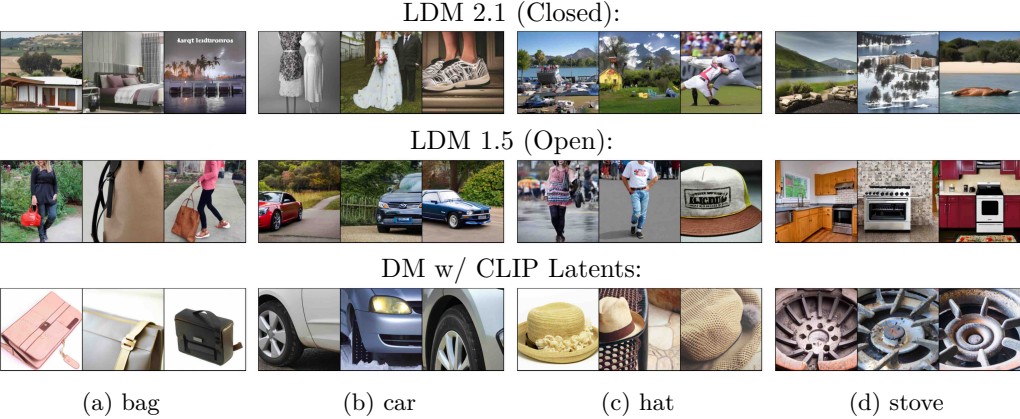

(a) bag  (b) car  (c) hat  (d) stove

Figure 12: Examples of images with CLIPscore in 10 percentile for different objects, regions and models. Images obtained by prompting with {object}. The CLIPscore is computed between the generated image and {object}.

We now report the consistency indicator using the prompts coming from the GeoDE dataset. Figure 11 displays bar plots representing the average across objects of the 10-percentile of CLIPScore for three model prompting setups. In the Appendix we report the full distribution of CLIPscores for each model/region setup.

**Observations when prompting without geographic information.** Figure 11a depicts CLIPScores for models prompted with `{object}`. We observe that LDM 2.1 (Closed) reaches the lowest consistency indicator suggesting that this model has some object-image consistency issues. This is confirmed via visual inspection: LDM 2.1 (Closed) frequently generates images that do not show the object named in the prompt (see first row in Figure 12). In addition, DM w/ CLIP Latents and LDM 1.5 (Open) achieve the highest consistency scores, with the latter reaching top scores. Our visual inspection of low-scored images for the two models suggests that both exhibit good object-generation consistency. However, the scenes generated by both models are very different, with LDM 1.5 (Open) generating objects in diverse backgrounds and DM w/ CLIP Latents producing object closeups with very simple, uniform backgrounds (see bottom two rows in Figure 12).

**Disparities when prompting with region.** Figure 11b depicts consistency scores for models prompted with `{object}` in `{region}`. First, we note that moving from `{object}` to `{object}` in `{region}` prompts reduces the consistency scores for all models. Images displayed in Figure 13 reveal that models rely less on the `{object}` part of the prompt and start to generate region specific content not necessarily related to the object, such as with *bag* for LDM 1.5 (Open) and LDM 2.1 (Closed) and *car in Africa* for DM w/ CLIP Latents. Next, we observe that DM w/ CLIP Latents achieves the highest consistency indicator. This is supported by visual inspection of Figure 13, where all its generations are consistent with the object. According to our consistency indicator, DM w/ CLIP Latents performs best for Europe and worst for Africa. LDM 2.1 (Closed) has the worst indicator score, achieving worst results for Europe and the best for the Americas. Indeed, the generations displayed in Figure 13 show low object consistency for this model for all regions. LDM 1.5 (Open) is second to the best performing model: for some objects and regions this model produces inconsistent generations, such as for *bag in Africa* and *hat in Europe*.

**Disparities when prompting with country.** Models prompted with `{object}` in `{country}` are depicted in Figure 11. In general, we observe that the consistency indicator is in line with our observations for the `{object}` in `{region}` setup – some model/region combinations benefit from more fine-grained geographic information while others are affected negatively. For example, we observe the largest positive change for generations of Africa by LDM 2.1 (Open) and the largest negative change for generations of the Americas by LDM 2.1 (Closed). Visual analysis of the images with low CLIPscore (Appendix Figure 22) reveals country specific content of the generations where objects such as *bag* frequently depict country's flag.

# 6 Limitations and Mitigation Directions

In this section we discuss limitations of our work and suggest directions for mitigations of observed disparities. We hope that this will inform future work focused on responsible text-to-image generative systems.

## 6.1 Limitations

While the indicators we introduce are useful in understanding and benchmarking performance disparities between geographic regions in text-to-image generative models, they have some limitations.

**Restrictions of the reference dataset** First, the precision- and coverage-based indicators are dependent on the particular reference dataset of real images used for evaluation and are susceptible to their representation of groups. Thus, they may lead to skewed conclusions about quality and diversity if the real reference dataset is unreliable. In our work, we aim to address this by selecting two datasets collected independently across geographic regions and observe that findings are sensible when accounting for the representation of groups in the real dataset. For example, DollarStreet was collected with a special focus on rarely-photographed or low-income households and tends to show more variation across region than GeoDE. In Appendix A.5 we demonstrate that, unsurprisingly, results of our Indicators show that the biased generations are more representative of the images of Africa in DollarStreet than in GeoDE.

Dataset size is also an important consideration when using these metrics. A larger reference dataset can make it to challenging to cover all modes of variation while a smaller reference dataset may make it hard to estimate

LDM 2.1 (Closed):

LDM 1.5 (Open):

DM w/ CLIP Latents:

LDM 2.1 (Closed):

LDM 1.5 (Open):

DM w/ CLIP Latents:

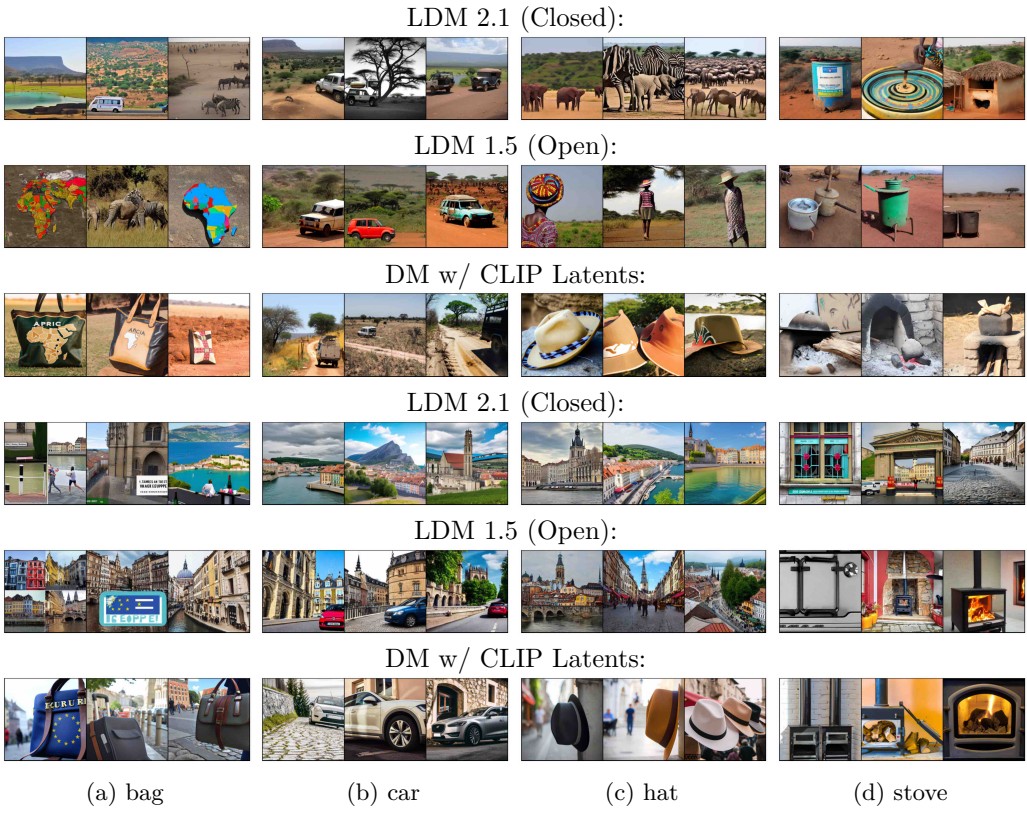

(a) bag          (b) car          (c) hat          (d) stove

Figure 13: Examples of images with CLIPscore in 10th percentile for different objects, regions and models. Images obtained by prompting with `{object} in {region}`. Rows 1-3 depict Africa and rows 4-6 depict Europe. The CLIPscore is computed between the generated image and `{object}`.

the manifold of real images. We also urge careful consideration of the portrayal of groups, potential biases in the data collection process, and reliability of labels in any reference dataset utilized for these indicators.

**Potential biases of feature extractors** In addition, all three indicators are dependent on the feature extractor used. While the use of Inception V3 is well-established when assessing performance of image generative models (Heusel et al., 2017a; Casanova et al., 2021; Ramesh et al., 2022; Rombach et al., 2021), future work may investigate the effect of different extractors on observed disparities. Prior work has shown that the InceptionV3 model can be susceptible to "fringe features" appearing in ImageNet (Kynkäänniemi et al., 2023) and have a bias towards texture over shape (Geirhos et al., 2022). In addition, the lack of geographically diverse images in ImageNet (Shankar et al., 2017) may mean the InceptionV3 model is better suited for identifying similarities of Western-centric representations. More modern models trained on larger scale data, such as DINOv2 (Oquab et al., 2023), may be stronger candidates as feature extractors (Stein et al., 2023) even as their relative recency means that their biases are less explored.

For the object consistency indicator we aim to reduce possible impact of group-level biases in CLIPscore by using only the `{object}` term when measuring similarity. However, the indicator would be impacted if CLIP finds certain variations of a given object (such as a large stove top accompanied by an oven) more representative of the concept than others (like a camp stove or a portable electric stove top). A future study analyzing how various feature extractors relate to human preference of quality, diversity, and consistency across geographies would provide additional guidance regarding the operationalization of these metrics.

**Not able to replace qualitative evaluations** While these measurements allow the identification and benchmarking of trends in performance across regions without relying on costly and time-consuming human evaluations, they should not replace thorough qualitative analyses that allow understanding of *why* and

*how* observed disparities occur. Furthermore, these assessments are not sufficient for understanding more fine-grained patterns such as imbalances *within* regions, *e.g.* by income-level, or patterns of co-occurrences between multiple object classes in a single image.

**Possible extensibility to other demographic groups**   We hope that future work investigates the application of this method to different types of demographic groups with the use of a reference dataset containing demographic information like gender, age, and skin-tone and target classes.

### 6.2   Mitigation Directions

Our analysis highlights the need for future work focused on developing methods for reducing gaps in quality, diversity, and consistency across geographic groups. For the models trained with publicly accessible data, it would be helpful to understand if similarly stereotypical and homogeneous images are present in the training data. For example, perhaps generated images reflect training data in which image-text pairs containing region information feature colorful and vibrant geographies common in travel websites or stock photos. In addition, mitigation efforts may focus on the role of the text-encoder in perpetuating geographic biases, as suggested by previous work (Gustafson et al., 2023; Richards et al., 2023). Finally, the variations in prompting may also be used to improve generation performance between groups. For example, markedness in language (Blodgett et al., 2021) of geographic information may affect model performance. Markedness refers to the fact that people mention certain kinds of information only when it is unusual, relevant, or remarkable. For instance, people from places with yellow bananas mention the color of a banana more often when it is green (Sedivy, 2003). It is possible that our prompts mark geographical information that speakers generally wouldn't, i.e. the default assumption would probably be that an object is wherever you are and thus geography may not be worth mentioning. In addition, prompting with languages beyond English may increase representation for regions where other languages are common, especially as multi-lingual generation capabilities become more prevalent. We discuss preliminary investigations in several of these areas in Appendix A.6.

## 7   Conclusion

In this work, we introduce three indicators used for evaluating disparities in realism, diversity, and consistency of generated images across geographic regions. These indicators complement existing time- and cost-intensive qualitative evaluation methods by allowing for the automatic benchmarking of models grounded in real world, representative datasets. They provide understanding of disparities in generations at increasingly granular levels of evaluation, aggregated first across objects for each region, then split across objects and regions. Furthermore, they allow the disentangling of cases when models struggle to generate images that are consistent to a given prompt from those in which generations lack realism and diversity while also accounting for variations in density of representations between regions.

We use these indicators to evaluate images of objects generated by widely used state-of-the-art text-to-image models across regions, with prompts of increasing geographic specificity. Through the Region Indicator, we find that existing models often have less realism and diversity of generations of objects when prompting with Africa and West Asia than Europe. This is reflected in qualitative analyses, where, *e.g.* we see the prompt *car in Africa* yields homogenous, boxy SUVs in consistently desert-like backgrounds. With the Object-Region Indicator, we see that region-level disparities occur more for some objects than others. For example, we find that generations of *stove* tend to have higher realism for Europe than Africa and West Asia. In some cases, disparities are due to differences in representation of the object in the real world reference dataset, but more often they are due to stereotypes embedded in text-to-image systems. With the Object Consistency Indicator, we find that prompting with geographic information can come at a cost to object consistency. Specifically, models like LDM 2.1 (Closed) struggle to generate images of a given object when prompts include geographic information. Of notable concern, we see from all three indicators that progress in image quality has come at the cost of real-world geographic representation. Altogether, these analyses demonstrate the indicators' efficacy in measuring limitations in the representation diversity of generated images.

We hope that this work allows for a more thorough understanding of geographic disparities in existing text-to-image generative models and encourages frequent benchmarking when iterating on model development.

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

# A  Appendix

## A.1  Reference Datasets

Figure 14 shows example images from each of the reference datasets (GeoDE Ramaswamy et al. (2023) and DollarStreet Rojas et al. (2022)) used for our Indicators.

While DollarStreet was formally recognized as a computer vision dataset in 2022 (Rojas et al., 2022), it has been used extensively for evaluations of geographic biases since 2019 (DeVries et al., 2019a; Singh et al., 2022; Goyal et al., 2022; Hall et al., 2023a; Gustafson et al., 2023; Richards et al., 2023).

## A.2  Additional Details: Region Indicator

Figure 15 shows the precision and coverage measurements for models evaluated with DollarStreet as the reference dataset. As discussed in the main text, prompting with regions shows an *improvement* for Africa. We believe this is due to Dollar Street's focus on low-income, remote locations, as well as its lack of enforcement of a minimum size for pictured objects.

## A.3  Additional Details: Region-Object Indicator

Figures 16 show the average object-region indicator for `{object} in {region}` and `{object} in {country}`. As shown in the figure, most models benefit from leveraging country information for West Asia and Africa. These benefits are especially pronounced for LDM 1.5 (Open) and LDM 2.1 (Open), whereas GLIDE only benefits modestly for Africa and East Asia.

## A.4  Additional Details: Object Consistency Indicator

To accompany our discussion of CLIPScore measurements in the main text, we show the box plots representing the CLIPScore across regions and objects in Figure 17.

## A.5  Additional Example Images

In this section, we show additional example images.

Figure 18 shows generated images corresponding to the prompt `{object}` for DM w/ CLIP Latents and LDM 1.5 (Open), as well as the GeoDE and DollarStreet reference images. Figure 19 shows images generated with the prompt `{object} in {region}`, and Figure 20 shows images generated with `{object} in {country}`.

To supplement the CLIPScore examples in the main text, we include examples of images with CLIPScore in the 10th percentile for East Asia using the prompt structure `{object} in {region}` in Figure 21. In Figure 22 we show example images in the 10th percentile across Africa, East Asia, and Europe using the prompt structure `{object} in {country}`.

## A.6  Preliminary Mitigation Analyses

We perform initial investigations into whether prompting with non-English languages prevalent in under-performing regions may help in reducing disparities. In this work we observed that LDM 1.5 (Open) and LDM 2.1 (Open) have stereotypical representations for the prompt "car in Africa" and "stove in Africa." In a small experiment, we prompt using Arabic, Hausa, and Zulu, which correspond to the most spoken non-English languages in the three African countries best represented in the GeoDE dataset (Egypt, Nigeria, and South Africa, respectively). We find that images generated with non-English languages tend to struggle with prompt consistency and continue to show stereotyped representations, as shown in Figure 23. Despite these initial observations, we believe this would be an interesting area to explore in more depth, especially as multi-lingual generation is increasingly supported.

We also perform a preliminary analysis of possible biases in LAION-2B-en, which is used in the training of LDM 1.5 (Open) and LDM 2.1 (Open). We perform a search of approximately 14.8 million image-text

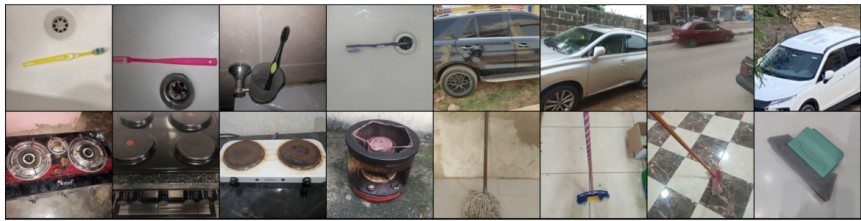

(a) Africa, GeoDE

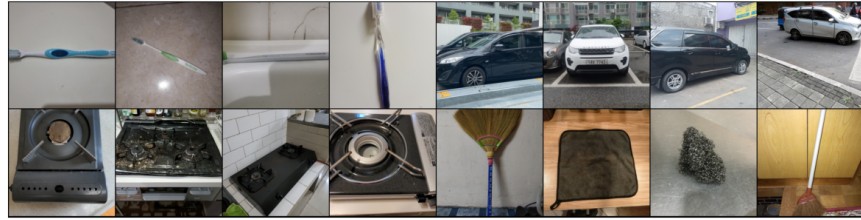

(b) East Asia, GeoDE

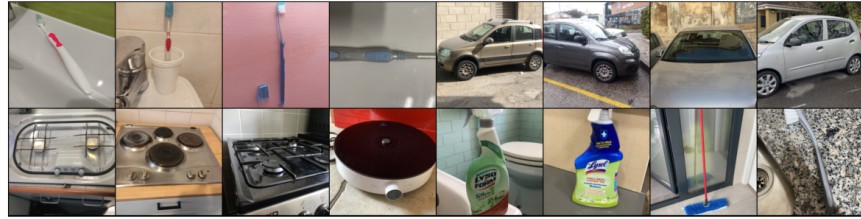

(c) Europe, GeoDE

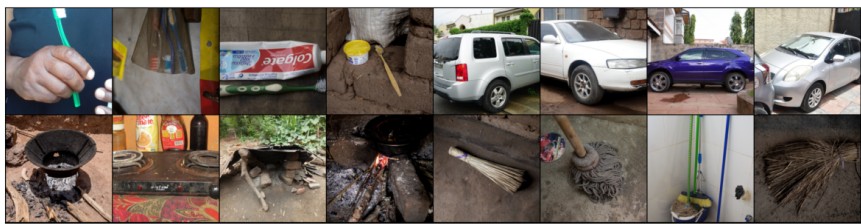

(d) Africa, DollarStreet

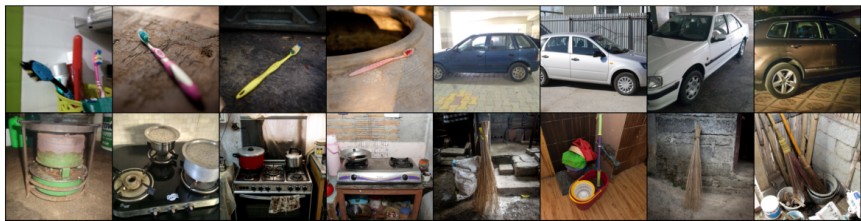

(e) Asia, DollarStreet

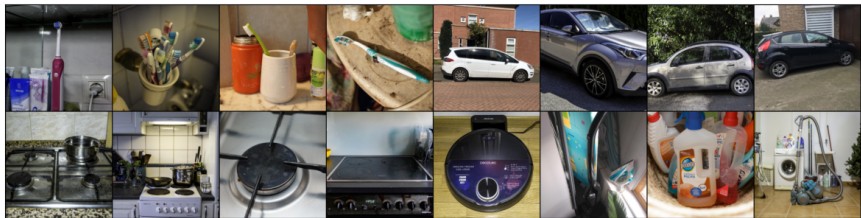

(f) Europe, DollarStreet

Figure 14: Examples of images from GeoDE (a, b and c) and DollarStreet (d, e, and f) datasets. For each dataset we display three regions and four random images of the following objects: toothbrush, car, stove and cleaning equipment.

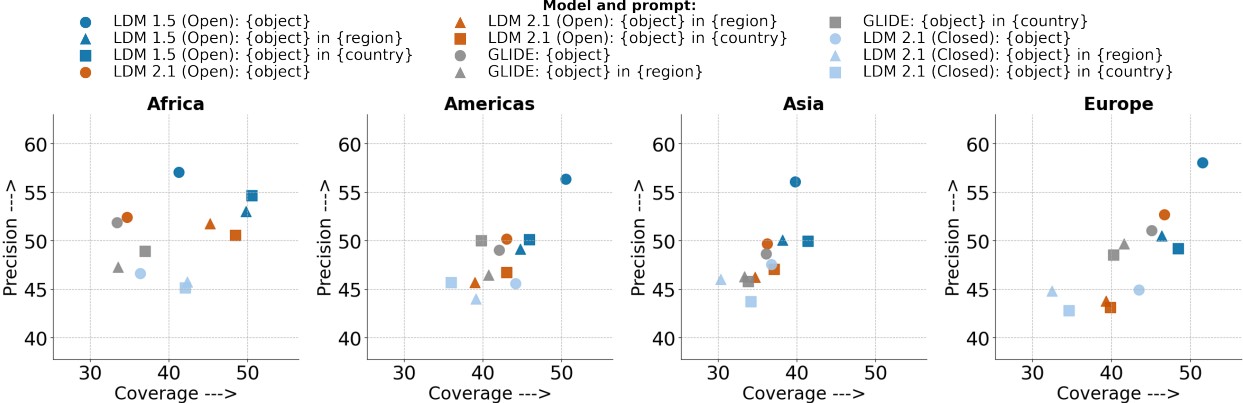

Figure 15: Precision and coverage measurements evaluated with the DollarStreet dataset.

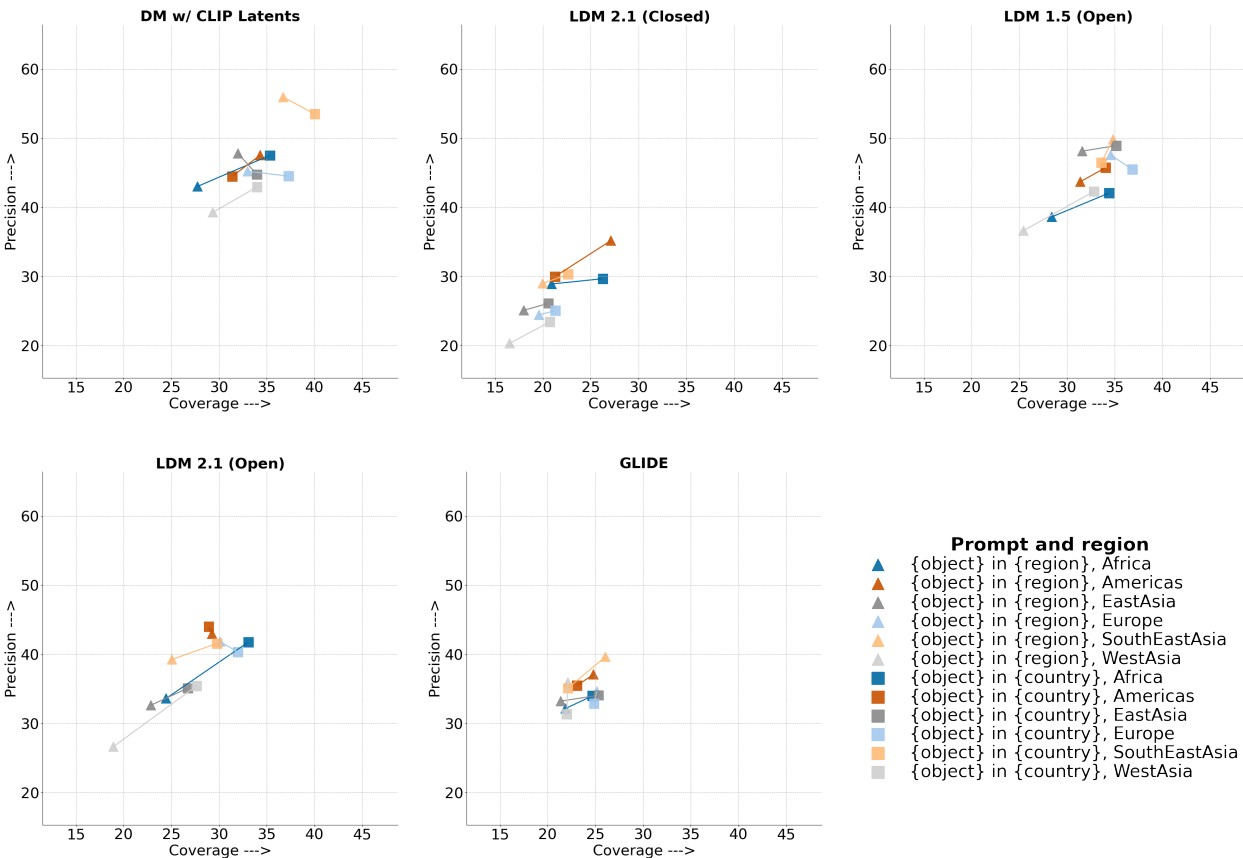

Figure 16: Average Region-Object Indicator for `{object} in {region}` and `{object} in {country}` prompt.

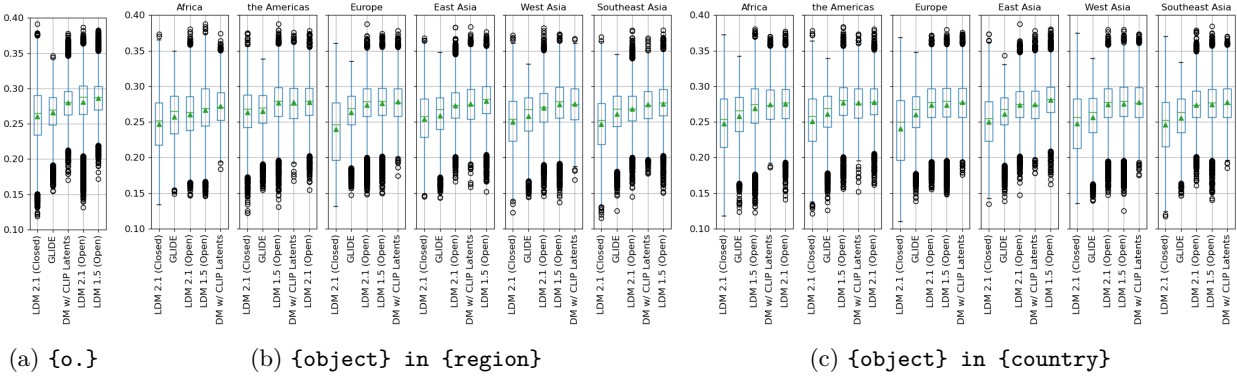

(a) {o.}  (b) {object} in {region}  (c) {object} in {country}

Figure 17: Average object CLIPScore in for images generated with GeoDE object prompts across various prompt structures, regions, and models. The box covers the range between Q1 and Q3 quartile values of the data. The whiskers extend up to 1.5 * IQR (IQR = Q3 - Q1) from the edges of the box. Outliers are plotted as separate dots. Box plots are sorted by avgerage score, indicated by the triangle.

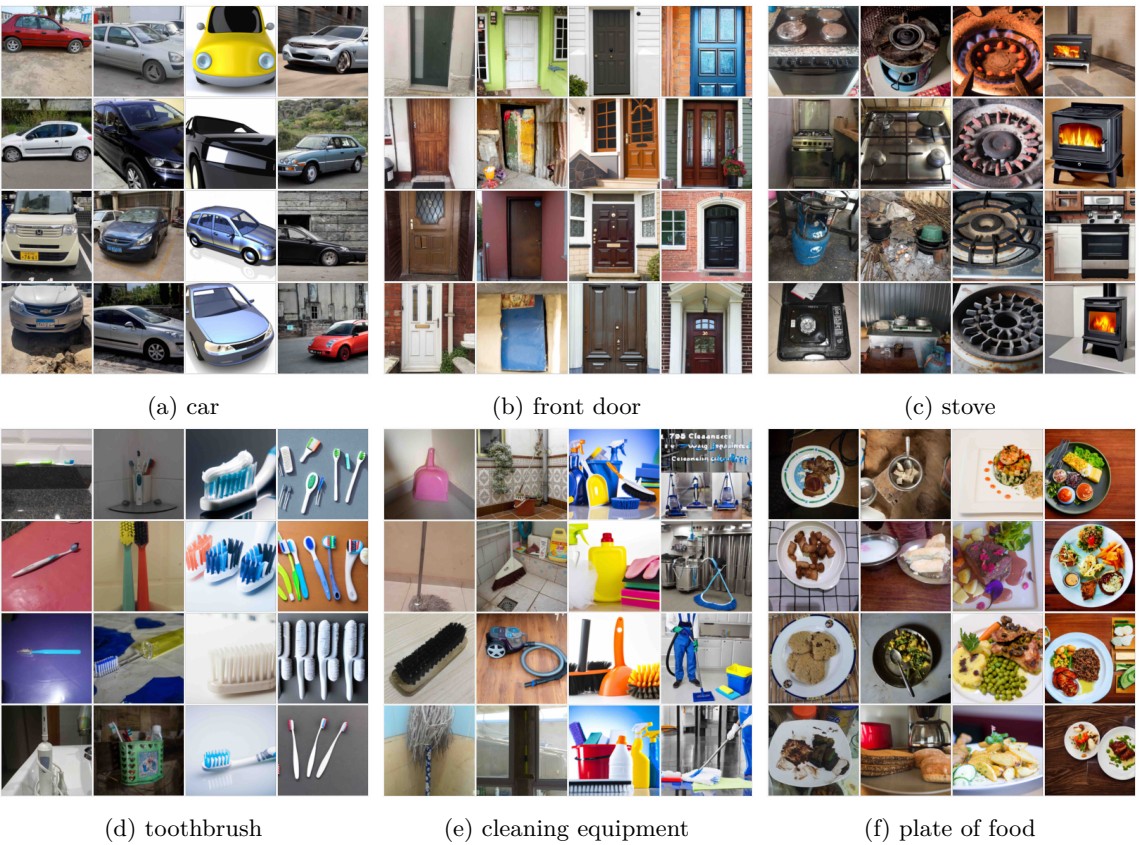

(a) car  (b) front door  (c) stove

(d) toothbrush  (e) cleaning equipment  (f) plate of food

Figure 18: Examples of generated images when prompting with {object}. Columns in each object-square correspond to GeoDE, DollarStreet, DM w/ CLIP Latents, and LDM 1.5 (Open).

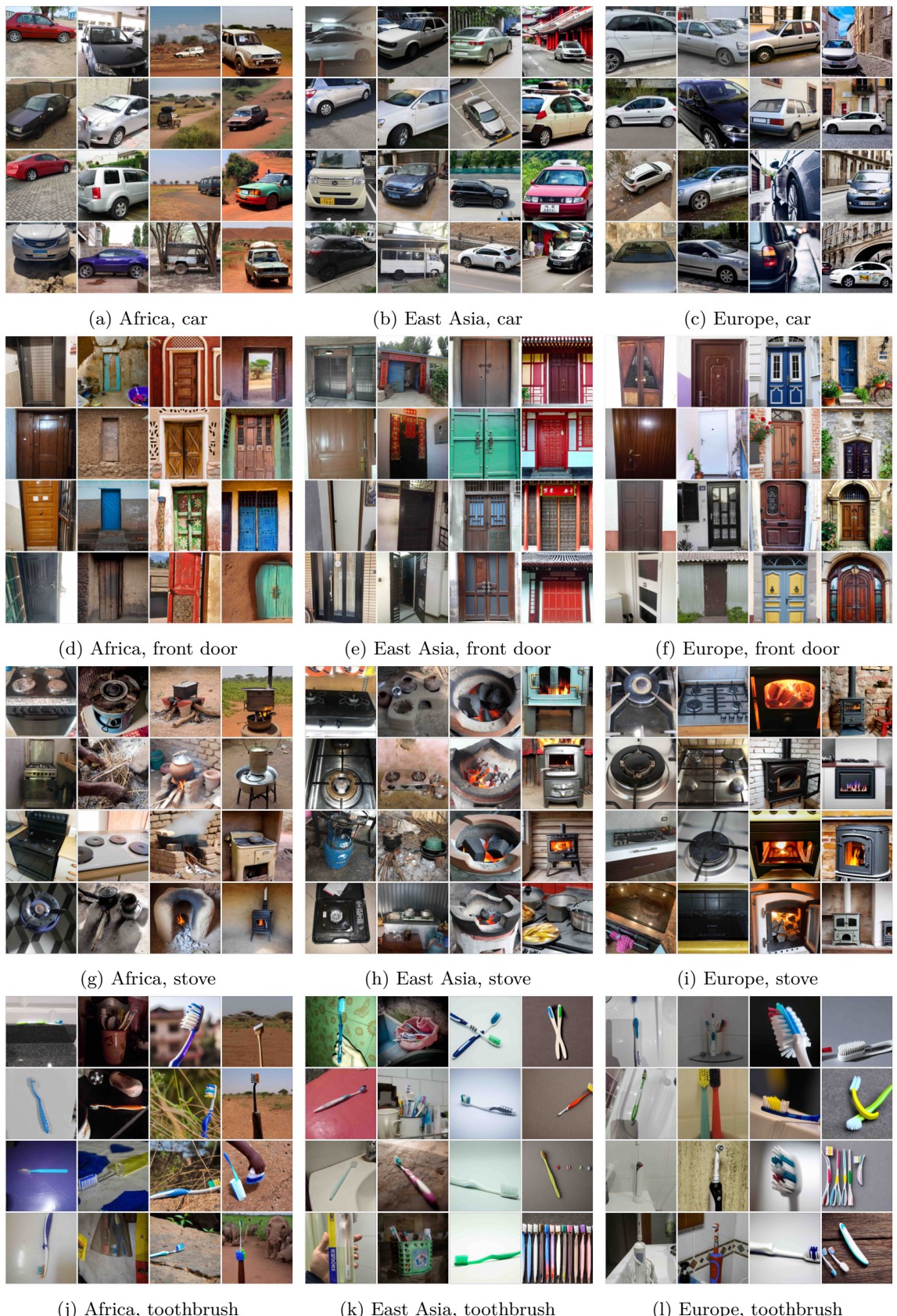

(a) Africa, car      (b) East Asia, car      (c) Europe, car

(d) Africa, front door      (e) East Asia, front door      (f) Europe, front door

(g) Africa, stove      (h) East Asia, stove      (i) Europe, stove

(j) Africa, toothbrush      (k) East Asia, toothbrush      (l) Europe, toothbrush

Figure 19: Examples of generated images when prompting with `{object} in {region}`. Columns in each object-square correspond to GeoDE, DollarStreet, DM w/ CLIP Latents, and LDM 1.5 (Open).

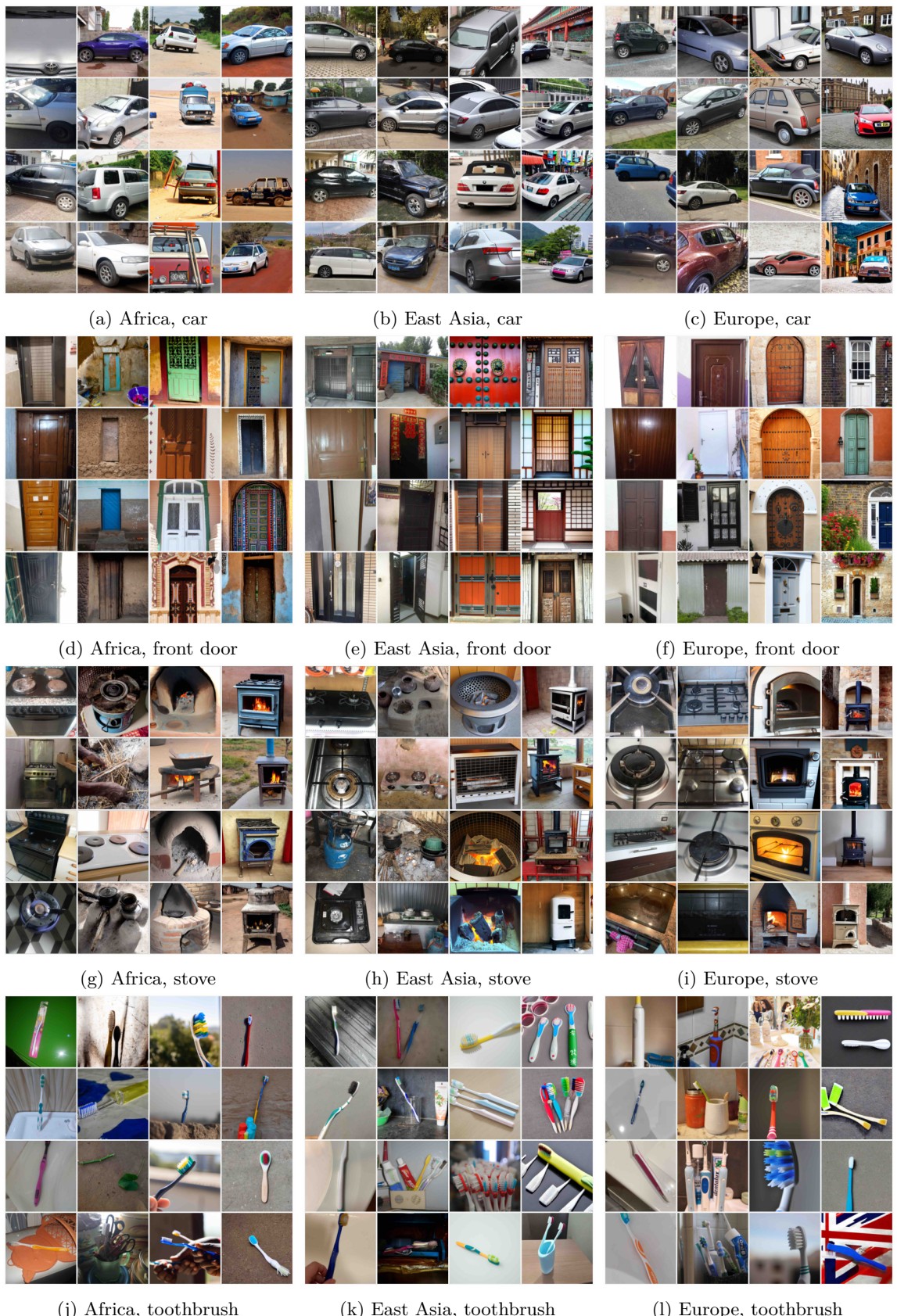

(a) Africa, car      (b) East Asia, car      (c) Europe, car

(d) Africa, front door      (e) East Asia, front door      (f) Europe, front door

(g) Africa, stove      (h) East Asia, stove      (i) Europe, stove

(j) Africa, toothbrush      (k) East Asia, toothbrush      (l) Europe, toothbrush

Figure 20: Examples of generated images when prompting with `{object} in {country}`. Columns in each object-square correspond to GeoDE, DollarStreet, DM w/ CLIP Latents, and LDM 1.5 (Open).

LDM 2.1 (Closed):

LDM 1.5 (Open):

DM w/ CLIP Latents:

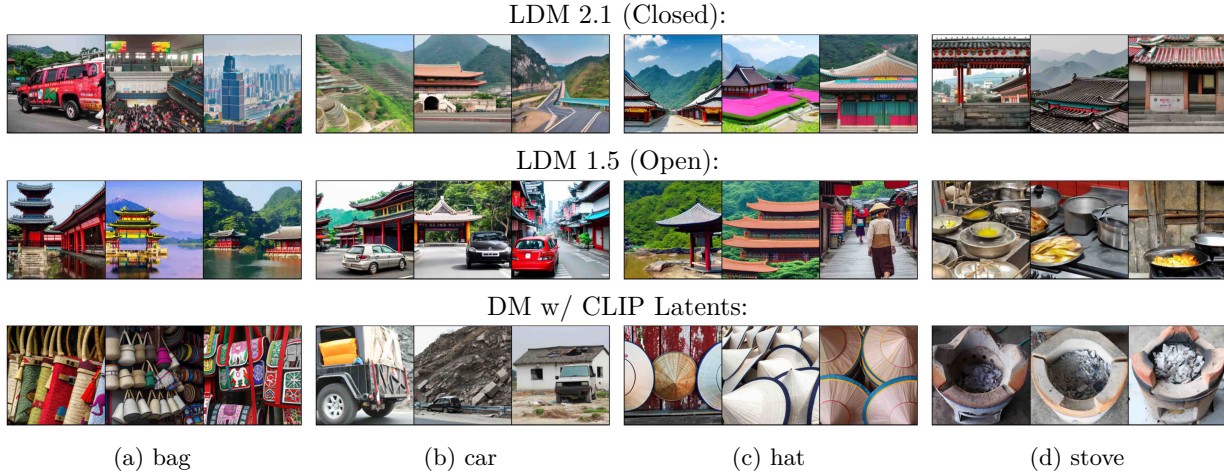

(a) bag (b) car (c) hat (d) stove

Figure 21: Examples of images with CLIPscore in 10th percentile for different objects and models for East Asia. Images obtained by prompting with `{object} in {region}`. The CLIPscore is computed between the generated image and `{object}`.

LDM 1.5 (Open):

DM w/ CLIP Latents:

LDM 1.5 (Open):

DM w/ CLIP Latents:

LDM 1.5 (Open):

DM w/ CLIP Latents:

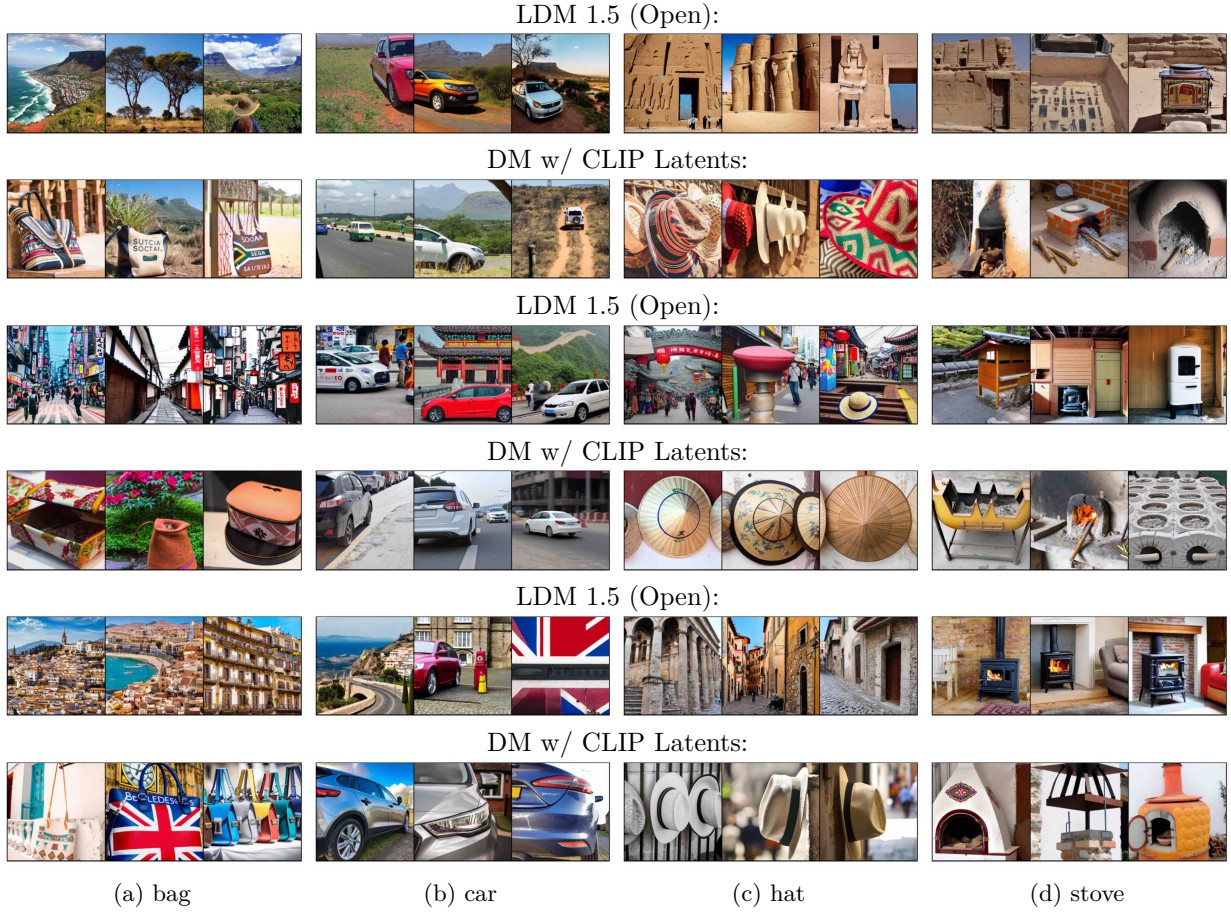

(a) bag (b) car (c) hat (d) stove

Figure 22: Examples of images with clip score in 10 percentile for different objects, regions and models. Images obtained by prompting with `{object} in {country}`. Rows 1-2 depict Africa, rows 3-4 depict East Asia, and rows 5-6 depict Europe. The CLIP score is computed between the generated image and `{object}`.

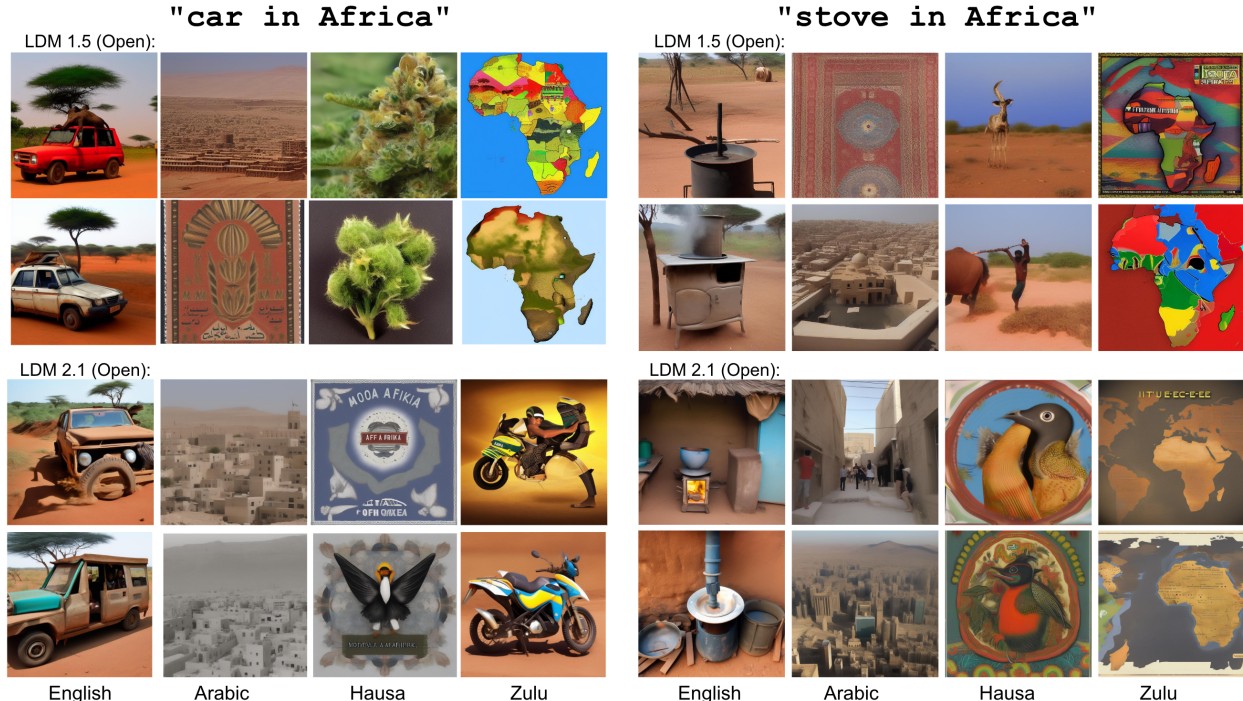

Figure 23: We prompt using Arabic, Hausa, and Zulu, which correspond to the most spoken non-English languages in the three African countries best represented in the GeoDE dataset (Egypt, Nigeria, and South Africa, respectively).

pairs to identify images whose captions contain either "car" or "bag" in conjunction with either "Europe" or "Africa". While fewer than 50 captions contain these terms, their associated images do provide some initial insights. For example, it seems that colorful and vibrant geographies are frequently conveyed in the images matching our keywords, similar to the generated images with the respective prompts. We also see that types of bags in LAION between the two regions differ, where images pertaining to Africa have more totes and images pertaining to Europe have more luxury-looking bags. In addition, there are fewer cars and bags depicted in the images of Africa that contain captions with those objects than in images pertaining to Europe. We include examples in Figure 24. We recommend more thorough analyses of the role that the training data plays in the disparities observed in our paper as an important area of future study.

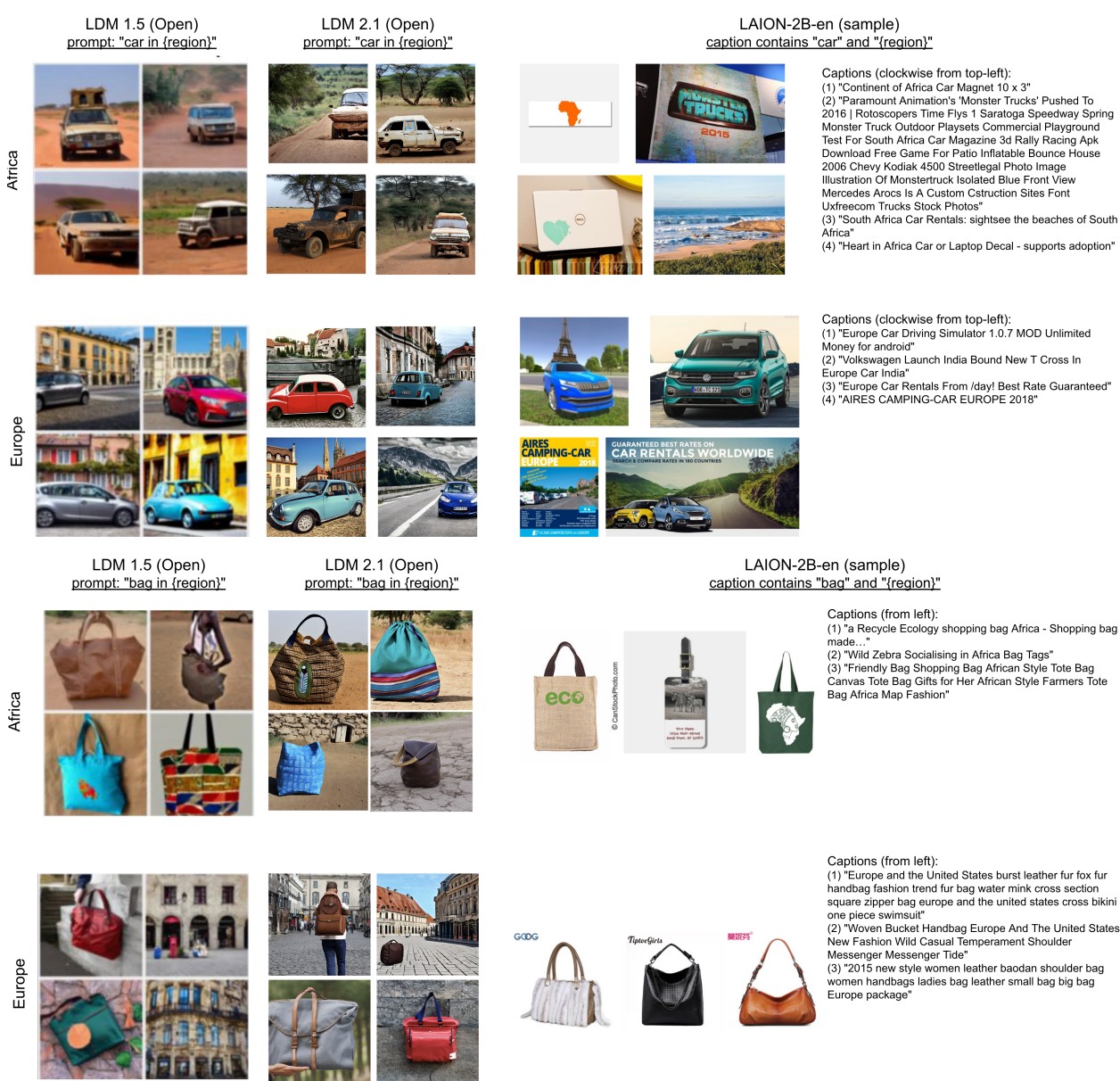

Figure 24: We perform a search of approximately 14.8 million image-text pairs to identify images whose captions contain either "car" or "bag" in conjunction with either "Europe" or "Africa".

