# OpenReview forum: "DIG In: Evaluating Disparities in Image Generations with Indicators for Geographic Diversity"
_TMLR — Accepted by TMLR_

### Review · Reviewer_uMRB · 2023-10-01

**Summary Of Contributions:**

This paper proposes a methodology for evaluating image generation models focusing on the biases that can arise across geographic regions/countries. The authors propose 3 metrics; the first two are computed with respect to a reference “groundtruth” set of real images, measuring precision, coverage and text-image similarity.  These metrics can be computed at the per-region level which allows the authors to compare performance disparity between two models on, e.g. Africa vs Europe.

Using these metrics, the paper studies several recent models including Dall-E 2, Stable Diffusion versions 1.5 and 2.0 and GLIDE.  Among other findings, the authors show that the more recent (closed/proprietary) SD 2.0 model, despite generating higher quality images, struggles to achieve good geographic coverage relative to the older SD 1.5 model.

**Audience:**

Yes

**Claims And Evidence:**

Yes

**Requested Changes:**

* The proposed precision and coverage measures are somewhat related to the more commonly used FID metric.  I would encourage the authors to add discussion comparing against FID.
* Precision and coverage also have some potential problems: For precision, if the reference dataset is too small, then there could easily be images generated that are realistic but not realistic by this measure. For coverage the tradeoff goes the other way --- if the reference dataset is too large, then it gets harder and harder to cover all the modes of variation --- would be good to add a discussion of this as well
* The reference datasets used in this paper are not that well known so comparing against them as the reference itself raises some questions.  Given that any interesting dataset that you could use would have some perspective/bias, it’s worth discussing the potential biases within these reference datasets (and provide summary statistics).
* (Optional but would be nice to have): Add Midjourney to the analysis
* Some nits:
  * I would recommend referring to models by their common names, which would improve readability
  * Section 4.4: the phrase “Next, we object-stratify the scores and compute the per class percentile 10 of CLIPscores” — this is hard to parse.

**Strengths And Weaknesses:**

Overall I found the manuscript to be well written and polished.  There are some interesting and timely findings (particularly about SD 1.5 vs 2.0).  And technically everything seems to be correct.  On the other hand, I would rate the technical novelty of this paper as rather incremental.  And the reference datasets used in this paper are not that well known (GeoDE, DollarStreet, both of which are relatively new and unestablished) so comparing against them as the reference itself raises some questions (more details below).

---

> ### Author Response · Authors · 2023-10-11
> **Author response**
>
> We thank the reviewer for their feedback about our work. We appreciate the reviewer finds our work well written, with relevant and interesting findings.
>
> We have submitted an update to our paper (changes noted in RED) to address recommendations from yourself and the other reviewers. We outline the full set of changes in our Official Comment on the main thread, but note here the changes that are especially relevant to your feedback:
> * In Section 4.1 we add a comment discussing why we pick precision, coverage, and CLIPScore in comparison to FID.
> * In our Limitations section, we add a section discussing the role of the reference dataset. This includes how varying representations between GeoDE and DollarStreet map to interesting findings with our Indicators and considerations of reference dataset size when using precision- and coverage-based metrics. We also include summary statistics about Dollar Street and GeoDE in Section 3.1. In particular, for GeoDE we have 27 objects with 180 images per object-region combination, and for DollarStreet we have 95 objects with approximately 3,500 images from Africa, 3,500 images from the Americas, 9,000 images from Asia, and 2,700 images from Europe. We also highlight Dollar Street's use in auditing vision models for geographic disparities since the publication of "Does Object Recognition Work for Everyone?" (2019) in the Appendix.
> * We clarify the phrase noted in Section 4.4.
>
> Once again, we appreciate your feedback. We remain available to address any additional comments from the reviewer.

---

> > ### Author Response · Authors · 2023-10-16
> >
> > Hi, we just wanted to check in to see if the reviewer had the chance to review our responses or had any follow-up questions / feedback. We are more than happy to continue iterating on your feedback for the remainder of the review period. Thank you!

---

### Review · Reviewer_5gg4 · 2023-10-02

**Summary Of Contributions:**

This paper studies potential geographic biases in prevalent diffusion-based text-to-image generation models. In particular, it proposes three indicators, Region Indicator, Region-Object Indicator and Object Consistency Indicator, and assesses five text-to-image models based on two existing geographically-balanced reference datasets. Several key findings are summarized from the results, highlighting severe geographic biases in all tested models.

**Audience:**

Yes

**Broader Impact Concerns:**

This paper is explicitly attempting to analyze biases and ethical issues in prevalent image generation models. It will have immense impact on the society.

**Claims And Evidence:**

Yes

**Requested Changes:**

I think the manuscript is worth acceptance, but adding some explanations addressing the weaknesses will improve its quality.

**Minor typos**
- Fig. 1: right subplot title: "Object-Region" -> "Region"

**Strengths And Weaknesses:**

**Strenths**

This is one of the first systematic quantitative studies of geographic biases in latest text-to-image generation models.
Despite the impressive visual quality of these T2I models, they are also notoriously known to be heavily biased towards certain modes.
A quantitative analysis of such biases based on geographics is pressing as these models become part of our everyday life, and this one arrives timely.

The studies are carefully conducted, including curating the datasets, designing the metrics and evaluating all models.
The findings are interesting and reveal severe geographic biases in all tested models:
- precision and coverage of the generated images are significantly worse in certain regions than others
- prompting with specific regions results in deteriorated performance, in terms of precision, coverage and consistency with prompt
- newer models trained with more data tend to perform worse in terms of geographic mitigating biases



**Weaknesses**

- The biggest limitation, as the paper also points out, is that the proposed metrics rely heavily on pre-trained feature extractors and well-balanced reference datasets, neither of which has proven sufficiently reliable. However, this is an extremely difficult problem, and these results are still extremely interesting and valuable at this point.
- In the Region-Object Indicator and Object Consistency Indicator, how was the particular list of objects selected for the study? They seem a bit arbitrary to me. What is the rationale behind this selection?
- The different indicator names are a bit confusing, especially compounded with various metrics. I wonder if there is a better way of structuring them. Fig. 1 also needs a better caption.
- It would be great if the authors can also briefly point to potential directions of addressing these biases for future research in the field of image generation.

---

> ### Author Response · Authors · 2023-10-11
> **Author response**
>
> We appreciate the reviewer's thoughtful comments regarding our submission. We also appreciate their recognition that the studies are carefully conducted with interesting and timely findings.
>
> We have submitted an update to our paper (changes noted in RED) to address recommendations from yourself and the other reviewers. We outline the full set of changes in our Official Comment on the main thread, but note here the changes that are especially relevant to your feedback:
> * We expand our Limitations section and make edits to more clearly delineate weaknesses of the feature extractor and how they might favor certain representations of objects. For example, we include an explanation of how the variation of representation between DollarStreet and GeoDE translates to different results and considerations of reference dataset size when using precision and coverage. We also more thoroughly discuss potential biases in the feature extractor, including towards features appearing in ImageNet, for texture over shape, and for geographic-specific representations present in its training dataset.
> * We clarify how we chose the object classes for analysis, explaining that these objects in highlight notable patterns of disparities that can be identified with the Indicator. Additional results covering all models and objects can be found in Appendix A.3.
> * We improve the caption for Figure 1 and update the right subplot title.
> * We introduce a new Mitigations section outlining potential directions for addressing the observed biases, informed by our qualitative analyses and feedback from other reviewers. These include an investigation into the training data and text encoder, using alternative languages when prompting, and studying the role of markedness when prompting.
>
> Once again, we appreciate your feedback. We remain available to address any additional comments from the reviewer.

---

> > ### Author Response · Authors · 2023-10-16
> >
> > Hi, we just wanted to check in to see if the reviewer had the chance to review our responses or had any follow-up questions / feedback. We are more than happy to continue iterating on your feedback for the remainder of the review period. Thank you!

---

### Review · Reviewer_2Aoq · 2023-10-02

**Summary Of Contributions:**

This work aims to study disparities in performance of text-to-image generative models when prompted to generate objects belonging to particular geographic locations.
In particular the authors with their experiments highlight that:

* Performance of a model prompted to generate a certain object class in a specific geographic location degrades significantly in terms of precision and coverage/diversity compared to the same model prompted to generate using just the object class.

* Among geographic locations some suffer way more from performance degradation and stereotypical representation than others. In particular “Africa” and “west/east Asia” suffer the most.

* Among the tested models newer and more powerful models seem to suffer more from this kind of disparities in performance depending on the geographic area considered.

The paper contributions are the analysis of the problem and the definition of the experimental methodology to quantify relevant metrics against a reference dataset of real images of objects in certain geographical locations. Datasets and metrics were already available in the literature, the contribution of the authors is the specific way of combining them together and the analysis of the results. Previous work studying this subject focused mostly on human evaluation that besides being costly and not scalable is in itself exposed to bias.

**Audience:**

Yes

**Broader Impact Concerns:**

No additional concerns.

**Claims And Evidence:**

Yes

**Requested Changes:**

Please have a look at the weaknesses listed above and in particular either consider extending the analysis to the training data (weakness 1) or propose/explore some mitigation strategies (weakness 2). Finally I would suggest improving the layout of the figures as detailed in weakness d.

**Strengths And Weaknesses:**

## Strengths

+ The topic considered is extremely interesting and as far as I know not explored extensively in the literature before.

+ The authors have been extremely detailed in detailing the experimental protocol and the choice made to compare models against each other. Replicating the findings should be straightforward.

+ I liked the effort of the authors to develop metrics that do not require human judgment to be able to scale the evaluation to many data and many models.

## Weaknesses

### Major

1. One point missing from the paper that I would have liked to see discussed more in details is what are the biases present in the training data used by these models and how much the final differences in the model performance depends from the training data used and how much by the architecture used (both in terms of number of parameters and objectives used to train them). This should be possible to assess at least for open source models trained on public data.
Most of these models are trained on web crawled image-caption pairs. By the nature of the way the data were collected I would expect geographic locations to appear in the caption only when elements of  the picture clearly refer to that country. Therefore in the training data geographic locations might appear only together with their “stereotypical” appearance and more powerful models might be more able to pick up on these nuances in the training data. I found that without discussing the training data perspective the analysis is missing the main source of the problem.

2. No solution or mitigation for the problem is proposed. This is not a weakness of the paper but it limits the contributions also in light of the not complete analysis (according to me) highlighted in weakness 1. In particular the paper highlights the existence of the problem and quantitatively measures it but it does not propose any mitigation strategy.


### Minor

3. The proposed metrics makes sense but have two flaws highlighted also by the authors: the dependance on a real reference dataset (that should be very large to guarantee stable metrics) and the dependance on a visual scoring model (Inception v3) that might have its own geographical bias. It would have been interesting to conduct a small user study to verify if the ranking of models emerging from the automatic metrics corresponds to the preferences expressed by human raters. This would have validated the results independently from the possible two flaws highlighted above and could have been limited to one or few prompt/models pairs.

4. I found the layout of the figures often confusing. For example in Fig. 3 in the first column “text-to-image models” corresponds to row, while for column 3-4-5 “models” corresponds to columns. I would suggest using fewer images at higher resolution and a consistent layout. For example in this specific case you could use “models” as a row, adding a row for the real images “GeoDE” and  add “no location/africa/europe/west Asia” as columns. The same comment can be applied to several images throughout the paper.

---

> ### Author Response · Authors · 2023-10-11
> **Author response**
>
> We thank the reviewer for the consideration of and feedback about our work. We appreciate their recognition that the topic is very interesting and that we present a detailed, replicable, and scalable experimental protocol.
>
> We have submitted an update to our paper (changes noted in RED) to address recommendations from yourself and the other reviewers. We outline the full set of changes in our Official Comment on the main thread, but note here the changes that are especially relevant to your feedback:
> * We add a new Mitigations discussion (Section 6.2), where we discuss the possible impact of training data used for the generative models on the observed biases. Following the reviewer's recommendation, we perform a preliminary analysis of possible biases in LAION-2B-en, which is used in the training of LDM 1.5 (Open) and LDM 2.1 (Open). We perform a search of approximately 14.8 million image-text pairs to identify images whose captions contain either "car" or "bag" in conjunction with either "Europe" or "Africa". While fewer than 50 captions contain these terms, their associated images do provide some initial insights. For example, it seems that colorful and vibrant geographies are frequently conveyed in the images matching our keywords, similar to the generated images with the respective prompts. We also see that types of bags in LAION between the two regions differ, where images pertaining to Africa have more totes and images pertaining to Europe have more luxury-looking bags. In addition, there are fewer cars and bags depicted in the images of Africa that contain captions with those objects than we do images pertaining to Europe. We include examples in the Supplementary material, titled "training_data_analysis_[car/bag]". While thorough investigations into the role of models in amplifying or preserving the biases in the training data is out of the scope of the presented analysis and the size of the LAION dataset makes it challenging to perform an in-depth analysis in the duration of our response period, we agree with the reviewer that more thorough analyses of the role that the training data plays in the disparities observed in our paper is an important area of future study. We include this recommendation in the new Mitigations section of our paper and add our analysis in Appendix A.6.
> * In our discussion of Mitigations (Section 6.2), we also highlight other areas for mitigation informed by our qualitative analyses and feedback from other reviewers. These include understanding the role of "markedness" in prompts, using alternative languages for prompting, and evaluating the role of various architectures.
> * We agree that a user study would be useful in validating the automatic metrics. While such a study is out of scope for this work, we add a recommendation for human-study analysis in our discussion of Limitations in Section 6.2.
> * We re-arrange Figure 3, 4, 9, and 10 to make the layout more consistent and readable.
>
> We reiterate our appreciation for your thoughtful comments and remain available to address any additional feedback from the reviewer.

---

> > ### Comment · Reviewer_2Aoq · 2023-10-13
> > **Thanks**
> >
> > Thanks for adressing my comments!
> > I think the revised version of the paper it's a clear improvement over the initial submission.

---

> > > ### Author Response · Authors · 2023-10-16
> > >
> > > Thank you for your consideration of our revisions!

---

### Review · Reviewer_XYe4 · 2023-10-02

**Summary Of Contributions:**

This work aims to measure geographic biases in generative models of images. It proposes a varity of metrics to characterise such biases, and uses these to perform a detailed empirical analysis of five text-to-image diffusion models, leveraging two existing geographically-unbiased datasets as references. It is found that overall, these models exhibit significant bias, with e.g. loss of realism and diversity when models are prompted to generate images depicting non-European regions.

**Audience:**

Yes

**Broader Impact Concerns:**

The paper does not raise any concerns in the traditional sense; however its purpose is to draw attention to such concerns arising with existing models, hence it should have a positive impact.

**Claims And Evidence:**

Yes

**Requested Changes:**

- not critical: unless there is some good reason, refer to models by the names people actually use (Stable Diffusion) etc. -- as-is it just seems to unnecessarily make life harder for the reader

- not critical: rearrange fig.3 (and other similar ones), since currently layout has models=rows at left but models=cols at top, which is rather confusing

See also the points above under 'weaknesses'.

**Strengths And Weaknesses:**

### Strengths:

The work is timely, with much attention given to image-synthesis models recently, yet little formal academic study of their biases.

The analysis is detailed and comprehensive. The different metrics are motivated, justified and described clearly; they capture various factors of interest. Relevant and diverse models are chosen for analysis. There is plenty of discussion and analysis to highlight interesting aspects of the results.

The paper is very clearly written and pleasant to read.

### Weaknesses:

Use of a single feature extractor (InceptionV3) in defining the metrics, without a proper analysis of that feature extractor's own bias, confounds almost all the quantitative results. Although the authors mention this in the conclusions, it is not sufficiently addressed. At very least, it should be possible to add a discussion of what possible effects biases in the feature-extractor (e.g. lack of African cars in its training data) would have on the different metrics.

There is no discussion of the impact of language, e.g. whether asking for an African car (or just a car) in an African language changes the pattern of results.

There is no discussion of how the relevant object classes are chosen for analysis -- e.g. are they those that exhibit most/least bias?

There is no attempt to measure training dataset bias -- e.g. for the LDMs trained on LAION, the data is public, and it should be possible to assess whether the model is more/less/equally biased as the data itself. This would increase the insight available from the paper, since currently it is unclear if these models are simply propagating training-set bias, or introducing bias of their own -- which presumably is important if we want to fix such biases.

---

> ### Author Response · Authors · 2023-10-11
> **Author response**
>
> We thank the reviewer for their thoughtful feedback about our work, and are happy to learn that the work is considered timely, well-motived, clear, and containing interesting analyses.
>
> We have submitted an update to our paper (changes noted in RED) to address recommendations from yourself and the other reviewers. We outline the full set of changes in our Official Comment on the main thread, but note here the changes that are especially relevant to your feedback:
> * In our Limitations section, we more clearly delineate weaknesses of the feature extractor and note how biases of InceptionV3's ImageNet training may affect the observations.
> * In our new Mitigations section, we suggest investigations into the role of the language used in prompting. Following the reviewer's recommendation, we also perform a small study into the viability of this method. In our paper we observe that LDM 1.5 (Open) and LDM 2.1 (Open) have stereotypical representations for the prompt "car in Africa" and "stove in Africa." We did a new test in which we prompt using Arabic, Hausa, and Zulu, which correspond to the most spoken non-English languages in the three African countries best represented in the GeoDE dataset (Egypt, Nigeria, and South Africa, respectively). We find that images generated with non-English languages tend to struggle with prompt consistency and continue to show stereotyped representations. We include examples in the Supplementary material, titled "languages_car_stove". Despite these initial observations, we agree with the reviewer's recommendation that this would be an interesting area to explore in more depth, especially as multi-lingual generation is increasingly supported, and include our analysis in Appendix A.6 to seed future investigations.
> * We clarify how we chose the object classes for analysis in Section 5.2, explaining that these objects in highlight notable patterns of disparities that can be identified with the Indicator. Additional results covering all models and objects can be found in Appendix A.3.
> * We recommend future directions of studying the impact of training data used for the generative models on the observed biases and perform a small study into related geographic bias in the LAION dataset. We perform a preliminary analysis of possible biases in LAION-2B-en, which is used in the training of LDM 1.5 (Open) and LDM 2.1 (Open). We perform a search of approximately 14.8 million image-text pairs to identify images whose captions contain either "car" or "bag" in conjunction with either "Europe" or "Africa". While fewer than 50 captions contain these terms, their associated images do provide some initial insights. For example, it seems that colorful and vibrant geographies are frequently conveyed in the images matching our keywords, similar to the generated images with the respective prompts. We also see that types of bags in LAION between the two regions differ, where images pertaining to Africa have more totes and images pertaining to Europe have more luxury-looking bags. In addition, there are fewer cars and bags depicted in the images of Africa that contain captions with those objects than we do images pertaining to Europe. We include examples in the Supplementary material, titled "training_data_analysis_[car/bag]". While the size of the LAION dataset makes it challenging to perform an in-depth analysis in the duration of our response period, we agree with the reviewers that more thorough analyses of the role that the training data plays in the disparities observed in our paper is an important area of future study. We include this recommendation in the new Mitigations section of our paper and add the analysis to Appendix A.6.
> * We re-arrange Figure 3, 4, 9, and 10 to make the layout more consistent and readable.
>
> Thank you again for your thoughtful comments, and we remain available to address any additional feedback from the reviewer.

---

> > ### Author Response · Authors · 2023-10-16
> >
> > Hi, we just wanted to check in to see if the reviewer had the chance to review our responses or had any follow-up questions / feedback. We are more than happy to continue iterating on your feedback for the remainder of the review period. Thank you!

---

### Author Response · Authors · 2023-10-11
**Summary of author responses (pt 1)**

We thank the reviewers for their thoughtful responses to our work. We appreciate that reviewers find our work to be "interesting and timely" as it is "one of the first systematic quantitative studies of geographic biases" and that "studies are carefully conducted" and "capture various factors of interest." Additionally, we are pleased that reviewers found the work to be "extremely detailed in detailing the experimental protocol" and that "replicating the findings should be straightforward."

We sincerely appreciate your recommendations for improvement to our work. Following your suggestions, we have implemented the following updates to our submission (noted in RED in the updated paper for easy identification):

(1) More thorough Limitations section...
We split the Limitations section (now Section 6.1) into easy-to-follow paragraphs and expand on important considerations discussed in your reviews. Our additions include:
* Restriction to reference datasets, including an explanation of how the variation of representation between DollarStreet and GeoDE translates to different results and considerations of reference dataset size when using precision and coverage. We also include summary statistics about dataset size in Section 3.1. Furthermore, we include in Appendix A.1 a reference to DollarStreet's history of use in illustrating biases in computer vision models extending from the influential work "Does Object Recognition Work for Everyone" introduced in 2019.
* More thorough discussions of potential biases in the feature extractor, including towards features appearing in ImageNet, for texture over shape, and for geographic-specific representations present in its training dataset.

(2) Addition of Root Cause Analysis / Mitigations section...
While we save thorough mitigation efforts for future work, we update the paper to include recommendations of potential mitigation directions in Section 6.2 based on our analyses. We also perform initial analyses as recommended by the reviewers. In particular, we:
* Perform a small test to understand whether alternative languages may help under-performing regions
In our paper we observe that LDM1.5 (Open) and LDM 2.1 (Open) have stereotypical representations for the prompt "car in Africa" and "stove in Africa." Following the reviewers' recommendations, we prompt using Arabic, Hausa, and Zulu, which correspond to the most spoken non-English languages in the three African countries best represented in the GeoDE dataset (Egypt, Nigeria, and South Africa, respectively). We find that images generated with non-English languages tend to struggle with prompt consistency and continue to show stereotyped representations. We include examples in the Supplementary material, titled "languages_car_stove". Despite these initial observations, we agree with the reviewer's recommendation that this is an interesting area to explore in more depth, especially as multi-lingual generation is increasingly supported. We recommend this in the new Mitigations section of our paper and add analyses to App A.6.
* Analyze the training dataset of the models LDM 1.5 and LDM 2.1.
The focus of the manuscript is on measuring potential geographic disparities in state-of-the-art text-to-image generative models when used out-of-the-box, given the increasingly widespread use these models are enjoying. Investigations into the role of models in amplifying or preserving the biases in the training data is out of the scope of the presented analysis. However, as recommended by the reviewers, we perform a preliminary analysis of possible biases in LAION-2B-en, which is used in the training of LDM 1.5 (Open) and LDM 2.1 (Open). Unfortunately, the datasets used to train the rest of the models are not publicly available. We search 14.8 million image-text pairs to identify images whose captions contain either "car" or "bag" in conjunction with either "Europe" or "Africa". While fewer than 50 captions contain these terms, their associated images do provide some initial insights. For example, colorful and vibrant geographies are frequently conveyed in the images matching our keywords, similar to the generated images with the respective prompts. We also see that bags in LAION between the two regions differ, where images pertaining to Africa have more totes and images pertaining to Europe have more luxury-looking bags. In addition, there are fewer cars and bags depicted in the images containing captions with "Africa" than images pertaining to Europe. We include examples in the Supplementary material, titled "training_data_analysis_[car/bag]".  While the size of the LAION dataset makes it challenging to perform an in-depth analysis in the duration of our response period, we agree with the reviewers that a thorough analyses of how training data affects disparities observed in our paper is an important area of future study. We recommend this in the new Mitigations section of our paper and add our analysis in App A.6.

(see Pt 2)

---

> ### Author Response · Authors · 2023-10-11
> **Summary of author responses (pt 2)**
>
> (continued from Pt 1)
>
> (3) Changes to formatting and clarifications.
> Finally, we make updates to the paper to address points of clarification and readability from the reviewers:
> * Unify layouts of Figures 3, 4, 9, and 10 and update captions to improve readability. Now, columns correspond to models/datasets and rows correspond to regions.
> * Improve caption of Figure 1 to be more descriptive and fix the right sub-plot title.
> * Add a comment to Section 4.1 discussing why we pick precision, coverage, and CLIPScore as opposed to FID.
> * Add a comment to Section 5.2 explaining that we select the specific subset of objects for our analysis as they highlight notable patterns of disparities that can be identified with this Indicator. Additional results covering all models and objects can be found in the Appendix.
>
> Thanks to the reviewers’ insightful suggestions, we believe these improvements have bolstered our analysis and enhanced the clarity of our work. We’re confident this work would be a valuable contribution to the research community.
>
> We remain available for further discussions and to answer any questions.

---

### Author Response · Authors · 2023-11-28

We thank the reviewers again for their thoughtful consideration of our work. We wanted to follow-up to see if there is any additional information needed from us and when we may expect to learn of a decision.
Thank you!

---

### Decision · Action_Editor_37qj · 2023-11-21

**Recommendation:** Accept as is

**Comment:**

All reviewers recommended acceptance. This is an interesting and different paper, reviewers believe it is a timely study and also are convinced about the results.

**Audience:**

Yes, reviewers believe that this work is timely in that generative models are proliferating and not much study has been made on their potential biases.

**Claims And Evidence:**

This work aims to measure geographic biases in generative models of images. It proposes a varity of metrics to characterise such biases, and uses these to perform a detailed empirical analysis of five text-to-image diffusion models, leveraging two existing geographically-unbiased datasets as references. It is found that overall, these models exhibit significant bias, with e.g. loss of realism and diversity when models are prompted to generate images depicting non-European regions. The evidence seems to support the conclusions.